## Review Article

# Longitudinal big biological data in the AI era

Adil Mardinoglu [1,2]✉, Hasan Turkez[3], Minho Shong [4], Vishnuvardhan Pogunulu Srinivasulu [5], Jens Nielsen [6], Bernhard O Palsson [7,8], Leroy Hood [9,10] & Mathias Uhlen [1]

## Abstract

Generating longitudinal and multi-layered big biological data is crucial for effectively implementing artificial intelligence (AI) and systems biology approaches in characterising whole-body biological functions in health and complex disease states. Big biological data consists of multi-omics, clinical, wearable device, and imaging data, and information on diet, drugs, toxins, and other environmental factors. Given the significant advancements in omics technologies, human metabologenomics, and computational capabilities, several multi-omics studies are underway. Here, we first review the recent application of AI and systems biology in integrating and interpreting multi-omics data, highlighting their contributions to the creation of digital twins and the discovery of novel biomarkers and drug targets. Next, we review the multi-omics datasets generated worldwide to reveal interactions across multiple biological layers of information over time, which enhance precision health and medicine. Finally, we address the need to incorporate big biological data into clinical practice, supporting the development of a clinical decision support system essential for AI-driven hospitals and creating the foundation for an AI and systems biology-based healthcare model.

**Keywords** Longitudinal Multi-omics Data; Artificial Intelligence; Systems Biology; Digital Twins; Precision Medicine
**Subject Category** Computational Biology

## Introduction

Complex diseases result from dysregulation in human biological processes, and revealing the molecular mechanisms driving the disease initiation and progression can provide insights into their aetiology (Doran et al, 2021; Mardinoglu and Nielsen, 2012). Current medical practices, such as 'one-size-fits-all' approaches, do not consider personal differences, thereby limiting the number of individuals who can benefit from diagnostic tools and treatment strategies (Babu and Snyder, 2023). Hence, understanding the complexity of biological processes and their malfunction at a personalised level requires a multifaceted approach (Naylor and Chen, 2010). This approach should integrate artificial intelligence (AI), systems biology, and big biological data comprising multi-omics, clinical, wearable device and imaging data, and information on diet, medications, toxins and other environmental factors (Fig. 1). Such an interdisciplinary and multi-dimensional strategy may enable the discovery of new links between the biological processes and underlying molecular mechanisms of diseases, such as cancers, neurological, immunological, infectious, inflammatory, rare and respiratory diseases and metabolic conditions, including obesity, diabetes, hypertension, hyperinsulinaemia and dyslipidaemia (Hasin et al, 2017; Hyduke et al, 2013; Mardinoglu and Nielsen, 2012).

Rapid advances in omics technologies and improvements in computational tools have led to the generation of high-quality and cost-efficient multi-omics data, empowering precision health and precision medicine approaches (Karczewski and Snyder, 2018; Mardinoglu et al, 2013). Multi-omics data, including genomics, epigenomics, transcriptomics, proteomics, metabolomics, lipidomics and metagenomics, provide multi-layered insights into biological processes associated with health and complex diseases (Babu and Snyder, 2023; Palsson and Zengler, 2010). Metabologenomics enables the identification of the functional link between metabolomics and genomics by integrating multi-omics data and studying the genetic, behavioural and environmental factors influencing metabolism (Mardinoglu et al, 2018; Mardinoglu and Palsson, 2024). Multi-omics data should also be enhanced with information on diet, medications and toxins, as well as data from wearable devices and electronic health records (EHRs). Wearable devices continuously monitor physiological parameters, including calories burned, blood pressure, heart rate, physical activity levels and sleep patterns. In contrast, EHRs include longitudinal clinical data, physical measurements, clinical notes and medical images.

Systems biology provides an integrated and holistic approach to understand biological systems by combining experimental and computational biology, thereby enabling the deciphering of molecular mechanisms in complex biological processes (Yurkovich et al, 2020). The systems biology approach can leverage multi-omics data to study the critical function of specific molecular components within the broader context of biological processes (Doran et al, 2021; Lam et al, 2020). Systems biology facilitates

[1]Science for Life Laboratory, KTH - Royal Institute of Technology, Stockholm, Sweden. [2]Centre for Host-Microbiome Interactions, Faculty of Dentistry, Oral & Craniofacial Sciences, King's College London, London, United Kingdom. [3]Department of Medical Biology, Faculty of Medicine, Atatürk University, Erzurum 25240, Turkey. [4]Graduate School of Medical Science and Engineering, Korea Advanced Institute of Science and Technology, Daejeon, Republic of Korea. [5]Vizzhy Longevity Inc., Middletown, DE 19709, USA. [6]BioInnovation Institute, Copenhagen DK-2200, Denmark. [7]Department of Bioengineering, University of California, San Diego, La Jolla, CA, USA. [8]Novo Nordisk Foundation Center for Biosustainability, Technical University of Denmark, Kongens, Lyngby, Denmark. [9]Phenome Health, Seattle, WA, USA. [10]Institute for Systems Biology, Seattle, WA 98109, USA. ✉E-mail: adilm@scilifelab.se

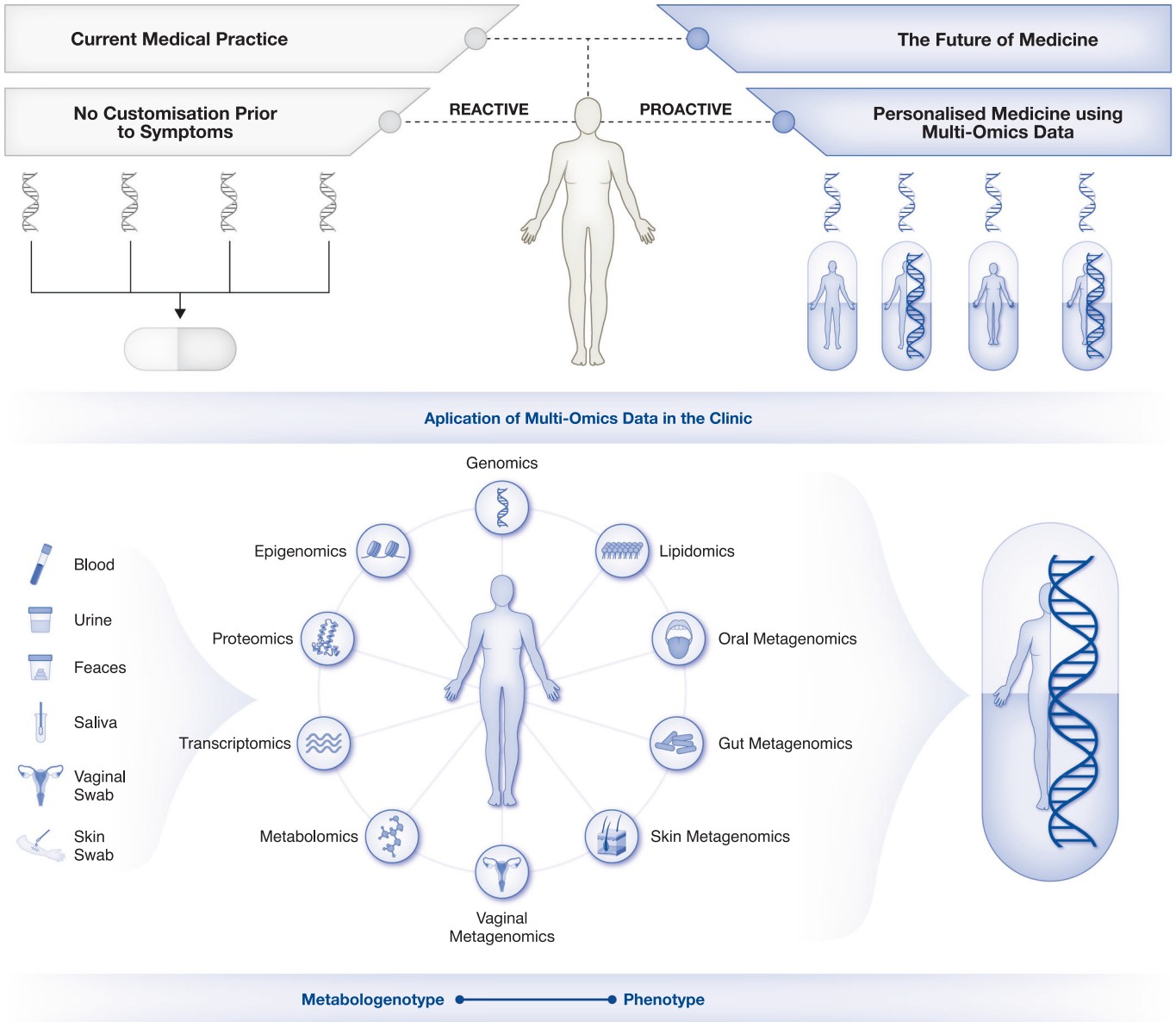

**Figure 1.  The future of medicine based on multi-omics data.**

Medicine is shifting from reactive, symptom-driven care to proactive, personalised approaches powered by multi-omics technologies. Current medical practice often relies on generalised treatments that address symptoms rather than root causes, overlooking individual genetic, molecular and environmental factors. The future of medicine leverages multi-omics to provide a comprehensive view of an individual's biology. This enables early disease prediction, prevention and the development of personalised therapies tailored to unique biological profiles, improving outcomes and minimising side effects. Critical advantages of multi-omics-based medicine include early detection of disease risks, tailored treatments, holistic health insights and reduced healthcare costs through preventive care. This data-driven approach is poised to revolutionise healthcare, transitioning from treating illness to maintaining and optimising health, heralding a new era of precision medicine.

multi-omics data analysis, integration, and interpretation by employing computational algorithms, biological networks and computational models at the genome-scale (Fig. 2). Within the context of specific medical questions, this practice is often referred to as systems medicine. This interdisciplinary field enhances our understanding of human biology, particularly by defining links between metabologenotypes, phenotypes, and environmental factors (Naylor and Chen, 2010).

AI plays a crucial role in utilising the vast amounts of big biological data to uncover the relationships between disparate data points and bridge the gap between big biological data and actionable biological insights. AI may facilitate a deeper understanding of disease mechanisms by distilling complex big biological data into discernible patterns. AI algorithms can be trained to recognise patterns and relationships within big biological data. For instance, AI has been successfully utilised in digital pathology, driven by the availability of large and structured datasets (Bera et al, 2019; Niazi et al, 2019; Song et al, 2023) and implemented in clinical settings to assist pathologists. Similarly, the successful application of AI in target discovery, target validation, and the

   

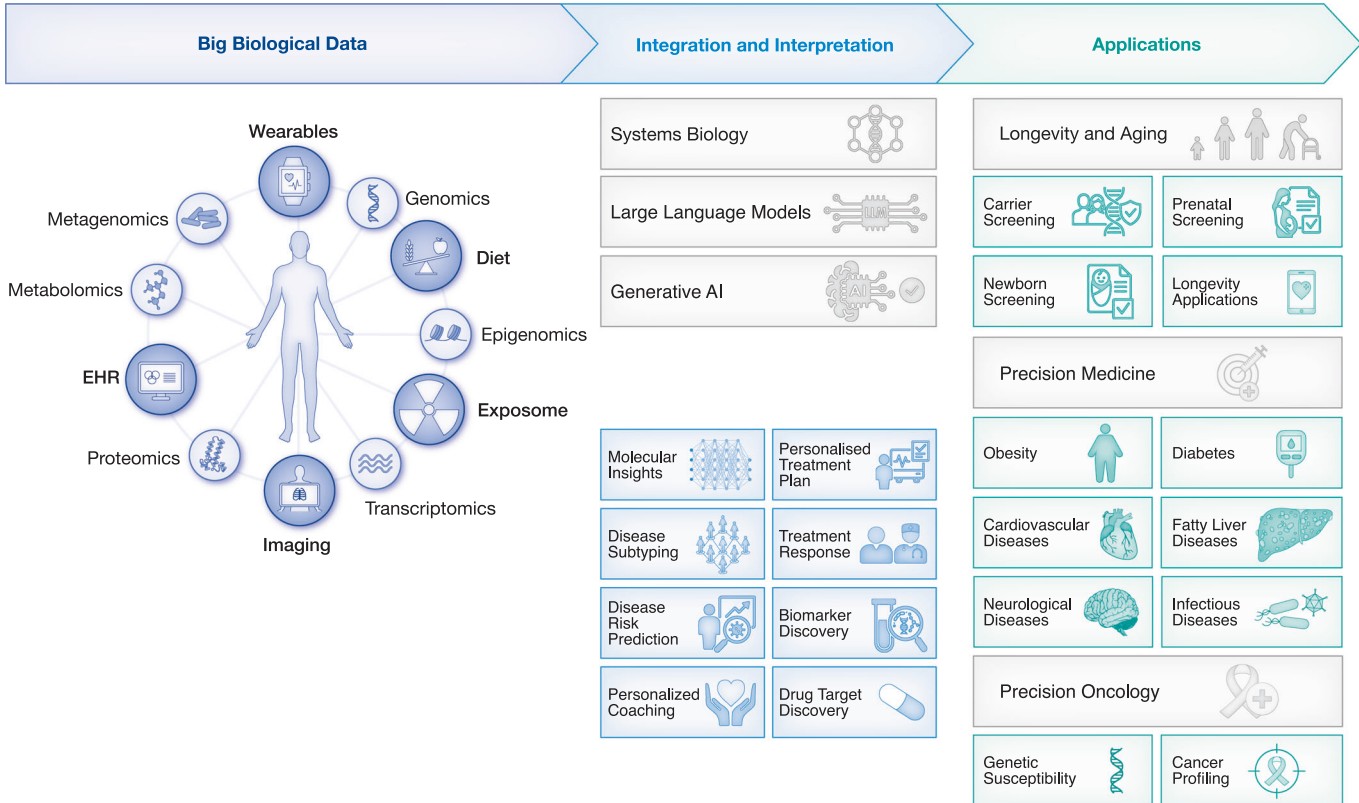

**Figure 2. The use of systems biology and AI for the integration of big biological data.**

Integrating and interpreting big biological data using systems biology and AI provides a transformative approach to healthcare, enabling time- and cost-efficient access to actionable insights. Combining the diverse data sources, including multi-omics, wearable devices, EHRs, imaging, and information on diet, drugs and toxins, with advanced tools, including biological networks, large language models and generative AI, may facilitate comprehensive analysis and interpretation. Applications include precision medicine for chronic and infectious diseases, precision oncology for cancer profiling, and interventions focused on longevity. Key outcomes include molecular insights, personalised treatment plans, disease subtyping, risk prediction, biomarker discovery, and drug target identification. This integrative framework aims to revolutionise healthcare, making it predictive, personalised and precise.

acceleration of drug development at both preclinical and clinical stages requires access to longitudinal big biological datasets.

Moreover, large language models driven by AI enable rapid biomedical text mining and enhance the interpretation of big biological data (Shandhi and Dunn, 2022). In this context, AI tools can identify connections between health, diseases, genes, transcripts, proteins, metabolites, lipids, microbiome, diet and toxins, as well as gain new insights into the complex biological systems involved in disease development and progression (Rajpurkar et al, 2022). Ultimately, the integration of AI, systems biology, disease-specific big biological data, and high-performance computing has the potential to develop effective treatment strategies by identifying novel biomarkers and drug targets and accelerating drug development (Bajwa et al, 2021).

Generating longitudinal big biological data is thus crucial for deciphering biological complexities in healthy individuals and patients with various pathological conditions and for developing innovative and effective therapies (Ho et al, 2020). Generating big biological data may also enable the personalised modelling of whole-body biological functions in real-time, more precisely by creating digital twins (Thiele et al, 2020). In this review, we first present recent examples of how systems biology and AI are utilised

to gain insights from multi-omics data and contribute to the development of effective treatments. Next, we examine the recently generated multi-omics data, which provides detailed information about health and disease states. Finally, we highlight the importance of generating longitudinal big data to comprehensively understand whole-body biological functions in both healthy individuals and patients with complex diseases, utilising digital twins. Such a multi-dimensional strategy may enable the creation of a clinical decision support system and ultimately lay the foundation for an AI- and systems biology-based healthcare system.

## Systems biology for the integration of multi-omics data

Human metabologenomics, which integrates metabolomics with genomics and other multi-omics data, including epigenomics, transcriptomics, proteomics, lipidomics and metagenomics, reveals critical molecular drivers involved in disease progression (Mardinoglu and Palsson, 2024). The advancement in systems biology has illuminated the onset and progression of complex diseases by generating biological networks and developing computational tools, which effectively analyse, integrate, and interpret multi-omics data (Mardinoglu and Palsson, 2024) (Fig. 2). Genome-scale models

(GEMs) serve as first principles-based tools for analysing and predicting the behaviour of complex metabolic networks, which are essential for sustaining life (Gu et al, 2019; Oberhardt et al, 2009). The first genome-scale reconstructions of the human metabolic network, Recon1 (Duarte et al, 2007) and EHMN (Ma et al, 2007), were introduced in 2007. Since then, several updates and expansions in the global human GEMs, including HMR1 (Agren et al, 2012), HMR2 (Mardinoglu et al, 2014), Recon2 (Quek et al, 2014; Smallbone, 2014; Swainston et al, 2016), Recon3D (Brunk et al, 2018) and Human1 (Robinson et al, 2020), have been developed. These reconstructions comprehensively account for genes, their sequence variants, transcripts, proteins, and metabolites, as they include all known biochemical reactions catalysed by the enzymes and transporters encoded in the human genome. Global human GEMs serve as highly curated knowledge bases and enable the reconstruction of a wide range of cell- and tissue-specific GEMs by manual curation (Gille et al, 2010) or applying automated and semiautomated model generation pipelines (Agren et al, 2014; Becker and Palsson, 2008; Shlomi et al, 2008; Vlassis et al, 2014). For instance, liver tissue-specific GEMs (Blais et al, 2017; Gille et al, 2010; Jerby et al, 2010; Mardinoglu et al, 2014; Vlassis et al, 2014) represent the unique metabolic processes occurring in the liver and have been widely used for studying the altered biological functions involved in the progression of liver diseases.

Microbe-specific (Monk et al, 2017; Shoaie et al, 2013; Tanaka et al, 2013) and microbial community GEMs (Bidkhori et al, 2021; Heinken et al, 2023; Heinken et al, 2025; Seaver et al, 2021) have been reconstructed for studying the critical role of microbial communities within humans, such as the oral, gut, vaginal, and skin microbiomes in health and disease states as well as for revealing the interactions between the host and microbiome. These GEMs assisted in the discovery of microbiome-based therapeutic targets and the development of personalised microbiome-based treatments. Moreover, whole-body metabolic models (Thiele et al, 2020; Zaunseder et al) have been developed by integrating cell- and tissue-specific GEMs and microbial community GEMs to study whole-body metabolic responses under different conditions.

Recently, we summarised a decade of progress in developing GEMs and the opportunities in using GEMs to design effective treatments for human diseases (Mardinoglu and Palsson, 2024). We and others have also provided recent examples of how multi-omics data have been analysed using GEMs to uncover the molecular mechanisms underlying metabolic diseases and to identify drug targets and biomarkers (Cortese et al, 2024; Mardinoglu and Palsson, 2024; Sen and Orešič, 2023).

Currently, GEMs capture the functions of ~20% of human genes. Integrating GEMs with other biological networks and generating integrated networks (INs) is essential. INs provide insights into the cellular functions represented by the remaining 80%, including DNA replication, protein synthesis, cell division, transcriptional regulation, signalling and protein–protein interactions in healthy and disease states (Mardinoglu and Nielsen, 2015). Considering that the generation of biological networks is foundational to systems biology, GEMs, protein–protein interaction networks (PPINs), signalling networks (SNs), gene regulatory networks (GRNs) and co-expression networks (CNs) have been combined to generate INs, and these networks have been continuously updated (Lee et al, 2016a; Lee et al, 2016b; Lee et al, 2017).

While GEMs have been instrumental in interpreting and simulating cellular metabolism across various contexts, systems biology offers a broader suite of methodologies for multi-omics data integration. GRN inference methods, including those based on transcription factor binding and epigenomic data, provide insight into the hierarchical control of gene expression and the impact of regulatory perturbations (Unger Avila et al, 2024). PPINs enrich this landscape by capturing the physical interactions among proteins, which are crucial for executing cellular functions and signal transduction (Ramos et al, 2024). SNs map the cascades of molecular interactions triggered by extracellular cues, allowing the dissection of dynamic cellular responses and pathway crosstalk (Garrido-Rodriguez et al, 2022). CNs, for instance, enable the identification of gene modules with correlated expression patterns, often revealing coordinated biological functions or shared regulatory control (Lee et al, 2017).

INs have been used to integrate and interpret multi-omics data, contributing to the development of translational medicine. Hence, systems biology provides insights into how biological systems function as a whole rather than focusing on individual components like genes or proteins in isolation through the analysis of biological networks. This interdisciplinary field enables the modelling of complex diseases, the identification of novel diagnostic biomarkers and drug targets, and, ultimately, the development of precision medicine.

While biological networks offer a mechanistic and interpretable framework for simulating metabolic phenotypes, alternative computational models provide complementary strengths in multi-omics data integration. Deep learning models, such as variational autoencoders and multi-modal neural networks, excel at capturing complex, nonlinear patterns across diverse omics layers and have been effectively used for patient stratification, feature extraction and disease prediction (Ballard et al, 2024; Ryu et al, 2018). Graph-based AI models, including graph neural networks (GNNs), are particularly suited for integrating biological networks (e.g. protein–protein interaction, gene regulatory or signalling networks) with omics data, enabling the modelling of topological dependencies and higher-order relationships (Valous et al, 2024). While often less interpretable than constraint-based models, these data-driven approaches offer scalability and flexibility, especially in high-dimensional, heterogeneous datasets. A comparative view that considers the trade-offs between mechanistic insight, predictive accuracy and interpretability can guide the choice of computational tools for different research and clinical objectives in precision medicine.

Additionally, probabilistic graphical models, including Bayesian networks, capture conditional dependencies among molecular entities, making them especially valuable for modelling causal relationships and integrating heterogeneous omics layers (e.g. transcriptomics with proteomics or metabolomics) (Jiang et al, 2025). These complementary approaches enable the construction of interpretable, data-driven models of biological systems and have been successfully applied in disease subtype classification, biomarker prioritisation and the discovery of novel mechanisms. Integrating diverse systems biology frameworks with AI-driven analytics may enhance our ability to decipher complex biological processes and foster a more comprehensive understanding of health and disease states.

## AI for the interpretation of multi-omics data

AlphaFold, an AI project launched in 2018, addresses the long-standing protein folding problem by predicting the

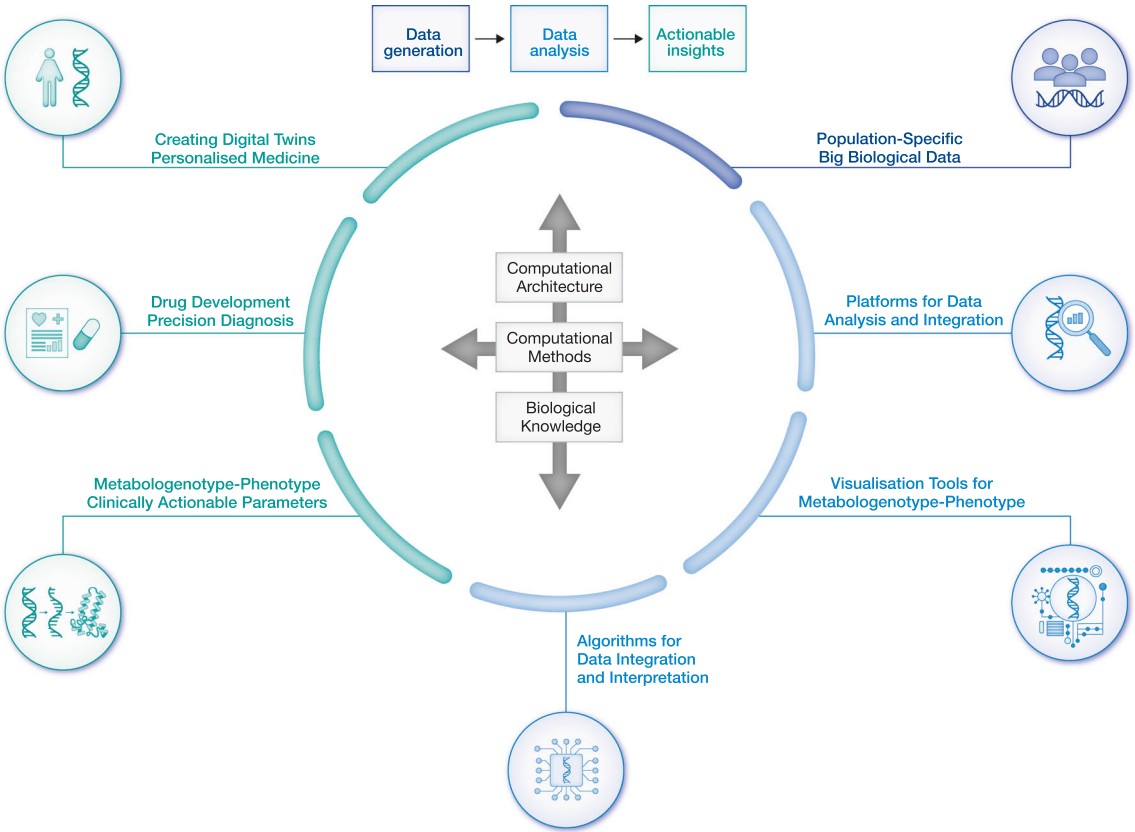

**Figure 3. Harnessing actionable insights from big biological data.**

This figure illustrates the transformation of multi-omics data into actionable clinical insights, highlighting its significance in personalised medicine, disease prevention and precision healthcare. Critical applications include tailoring treatments using multi-omics data for personalised medicine, enabling precision diagnosis and treatment by identifying clinically actionable biomarkers, uncovering metabologenotype and phenotype correlations to enhance understanding of disease mechanisms, and improving analysis accuracy by developing haplotype-aware algorithms. Genotype-phenotype mapping is supported by platforms such as ClinVar and ClinGen, while multi-omics platforms like VannoPortal and CAUSAL DB facilitate complex data integration. Population-specific datasets, including gnomAD and Genome Asia 100 K, provide diverse and inclusive insights. Advanced computational tools streamline data analysis, and rigorous validation through laboratory and clinical testing ensures the reliability of findings. This iterative process harnesses global datasets and cutting-edge computational methods to convert raw data into precise, actionable insights, revolutionising healthcare through predictive, personalised and preventive approaches.

three-dimensional structures of proteins from their amino acid sequences. AlphaFold uses deep learning and advanced algorithms to achieve atomic-level accuracy (Jumper et al, 2021). Its open-access AlphaFold Protein Structure Database provides over 200 million predicted structures, covering nearly all known proteins from sequenced genomes (Varadi et al, 2023). AlphaFold, with its cost-effective protein structure predictions, may drive advancements in disease research and drug discovery while aiding in the understanding of disease mechanisms and the design of novel proteins (Abramson et al, 2024; Nussinov et al, 2023).

AI is employed to analyse and integrate multi-omics data, leveraging fundamental biological and computational principles to uncover complex relationships across diverse datasets (Wei et al, 2023). By applying machine learning algorithms and advanced statistical models, AI can identify patterns and reveal insights that would be challenging to discern through conventional analysis methods (Fig. 3).

Earls et al. (Earls et al, 2019) examined biological age (BA), derived from molecular and physiological measurements, as a potential predictor of mortality and disease, surpassing chronological age

(CA) in accuracy. Longitudinal multi-omics data from 3558 individuals in a wellness programme were used to compute BA based on phenomics, genomics, proteomics, and metabolomics data. BA was higher than CA in individuals with chronic diseases, but participants in the wellness program showed a slower-than-expected rate of BA increase, suggesting that BA is modifiable and may indicate healthy ageing. Key predictors of BA included markers of metabolic health, inflammation, and toxin bioaccumulation, with BA changes aligning with positive and negative health conditions. Strong predictors of BA in clinical labs included glycated haemoglobin (HbA1c) for metabolic health, while proteomics identified agouti-related peptide (AgRP) as a significant negative predictor. Inflammation markers, such as CXCL9 and IL17D, were identified as positive BA predictors, whereas lymphocyte levels were determined as negative BA predictors. It has also been reported that environmental toxins, such as lead, mercury, and PFOS, have a strong influence on BA. These findings support BA as a valuable metric for monitoring ageing and potentially guiding personalised interventions.

Zeybel et al (Zeybel et al, 2022) conducted a longitudinal multi-omics study on 56 heterogeneous, well-phenotyped patients with

metabolic dysfunction associated steatotic liver disease (MASLD) to investigate the pathogenesis of hepatic steatosis (HS) accounting for the host–microbiome interactions. The study included comprehensive analyses of phenomics, proteomics, metabolomics, lipidomics and oral/gut metagenomics data to identify features associated with varying degrees of HS. Initially, the authors analysed each omics dataset and found that gut metagenomics data were the most predictive of HS degree, while inflammatory proteomics data were the least predictive of HS degree. The authors identified the top five predictive features from each omics dataset, including clinical, proteomics, metabolomics, oral metagenomics and gut metagenomics data and used these features to achieve the highest predictive score for the HS degree. Specifically, the top five predictive clinical features included alanine aminotransferase (ALT), aspartate aminotransferase (AST), uric acid, insulin and gamma-glutamyl transferase (GGT); metabolomics features included N,N-dimethyl-5-aminovalerate, 5-(galactosylhydroxy)-L-lysine, phenol glucuronide, N,N-dimethylalanine, and N-methyl-taurine; and proteomics features included CDCP1, CD244, LIF-R, FGF-21 and CXCL9. Notably, multi-omics classification significantly outperformed single-omics classification using all these selected features.

Watanabe et al (Watanabe et al, 2023) revealed the molecular changes associated with body mass index (BMI) by performing longitudinal multi-omics data analysis in a cohort of 1277 individuals enroled in the Arivale wellness program. The authors analysed 1111 blood analytes by generating phenomics, proteomics and metabolomics data to uncover the associations with BMI, genetic predispositions and gut microbiome composition. The authors predicted BMI based on these analytes using machine learning models and observed that their findings outperformed the currently used clinical measures. They also validated their findings with an external TwinsUK cohort (Moayyeri et al, 2013). The multi-omics analysis showed that changes in BMI influenced various omics measures differently in response to lifestyle interventions, with metabolomics-inferred BMI showing more significant reductions than actual BMI. The study identified distinct biomarker panels from each omics category (e.g. 62 metabolites and 30 proteins) as critical predictors for BMI, with the combined model (CombiBMI) achieving the highest predictive accuracy. Additionally, specific proteins (like leptin and FABP4) and other biomarkers emerged as solid indicators associated with BMI, though no single analyte matched the explanatory power of multi-omics-based models.

Meng et al. (Meng et al, 2024) conducted a longitudinal multi-omics study on 87 heterogeneous, well-phenotyped Alzheimer's disease (AD) patients to investigate the pathogenesis of AD, accounting for host–microbiome interactions. They generated phenomics, plasma proteomics, metabolomics, lipidomics and oral/gut metagenomics data to uncover molecular changes associated with cognitive functions in AD patients. Using multi-omics data, the authors identified critical features linked to the severity of cognitive decline in AD patients. AI-based models, including support vector machines (SVM), random forests (RF) and XGBoost, were applied to predict AD severity based on significant features from each omics analysis. The authors found that metabolomics and gut metagenomics data demonstrated high accuracy in single-omics classification. However, multi-omics classification using all three machine learning algorithms

outperformed single-omics predictions. Their integrative analysis highlighted the clinical relevance of specific plasma proteins (e.g. SKAP1 and NEFL), plasma metabolites (e.g. homovanillate and glutamate) and the gut microbiome species (e.g. Paraprevotella Clara) in predicting the severity of cognitive decline.

Jin et al. (Jin et al, 2025) performed a longitudinal multi-omics study involving 64 well-phenotyped Parkinson's disease (PD) patients, combining phenomics, plasma metabolomics, proteomics and oral/gut metagenomics data to investigate molecular mechanisms associated with PD severity, particularly motor dysfunction. The researchers identified a panel of 58 biomarkers, including clinical variables, proteins, metabolites, gut species and saliva species, providing more robust predictive performance for assessing PD severity than single-omics data alone. Machine learning models, including logistic regression (LR), SVM and RF, were used to validate the predictive power of these biomarkers. After feature selection, each omics dataset showed an improvement in prediction accuracy by about 10–30%, with a significant increase in AUC (0.15–0.4). The highest predictive accuracy was achieved when biomarkers from all omics datasets were integrated, with LR and SVM models yielding balanced accuracies of 87.75 and 86.28%, respectively, and AUC scores near 0.95. This multi-omics approach, mainly including metabolite biomarkers, showed strong potential in predicting PD severity.

These studies underscore the value of multi-omics analyses in capturing the complexity of disease-related biological changes. They also highlight the effectiveness of AI in integrating and interpreting multi-omics data to gain insights into complex disease mechanisms. These findings may have significant implications for the development of diagnostic and therapeutic approaches.

Large language models (LLMs), such as GPT, Llama, Gemini, BioBERT, and PubMedBERT, have recently emerged as powerful tools in biomedical informatics (Lu et al, 2024). These models, trained on large-scale biomedical corpora, can perform various tasks, including literature mining, clinical report summarisation and question answering, making them particularly useful for hypothesis generation and clinical decision support. For example, GPT-based models have been used to synthesise insights from vast biomedical literature, enabling the identification of novel gene-disease associations and mechanistic hypotheses (Wang et al, 2024). Similarly, BioBERT has shown strong performance in extracting biomedical entities and relationships, supporting the development of structured knowledge graphs that can be integrated with multi-omics data (Rehana et al, 2024). Recent efforts have also explored the coupling of LLMs with omics analysis pipelines to contextualise molecular findings within the current scientific landscape, aiding in interpreting results and prioritising biomarkers or drug targets (Toufiq et al, 2023). As LLMs evolve, their integration into precision medicine workflows is expected to enhance data interpretation, reduce information bottlenecks, and support more informed clinical decision-making.

## Population-scale initiatives for personalised big biological data

National Genome Projects have been initiated to identify the genetic architecture of populations living in more than 50 countries using whole-genome sequencing (WGS). These projects have been conducted to study genetic variants specific to a particular ancestry (e.g. a

fixed allele) or the distribution of rare variants in a population with frequencies of less than 1%. Bridging the gap between genomic and clinical practice and translating the outcomes of these projects into clinical benefits took many years, considering that the first draft of the human genome sequence was released in 2001 (Brlek et al, 2024; Giani et al, 2020). In clinical practice, genomic markers are used to screen, diagnose, and stratify patients, guiding tailored treatment strategies for rare diseases and cancers (Fig. 3).

The Human Protein Atlas (HPA) Project, launched in Sweden in 2003, aims to map all human proteins and transcripts across cells, tissues and organs using proteomics and transcriptomics and integrates all these data using AI and systems biology (Agaton et al, 2003; Uhlen et al, 2010). Advanced methods, including immunohistochemistry-based proteomics, RNA sequencing, mass spectrometry and spatial imaging, are employed to generate proteomics and transcriptomics data. This initiative provides an open-access, integrated multi-omics resource to advance understanding of human biology at health and disease states. Key features include (i) the Tissue Atlas (Uhlén et al, 2015), which catalogues protein and transcript expression in organs; (ii) the Subcell Atlas (Thul et al, 2017), which highlights protein localisation in subcellular compartments; (iii) the Cancer Atlas (Uhlen et al, 2017; Yuan et al, 2024), linking protein and transcript expression to cancers and survival of the patients; (iv) the Blood Atlas (Uhlén et al, 2019; Uhlen et al, 2019), detailing the functions of blood protein in health and disease states; (v) the Brain Atlas (Sjöstedt et al, 2020), focusing on the human brain's proteome and transcriptome; (vi) the Single Cell Atlas (Karlsson et al, 2021), which catalogues the transcript expression in single cells; (vii) the Cell Line Atlas (Jin et al, 2023), which catalogues the protein and transcript expression in different cell lines. There is also the Structure Atlas, which provides information about the structure of the protein and the Interactome Atlas, which provides a functional link between individual proteins and transcripts (Arif et al, 2021; Lee et al, 2018; Robinson et al, 2020). The HPA facilitates systems biology studies to discover drug targets and biomarkers and elucidates the underlying molecular mechanisms of diseases, making it an invaluable resource for translational research and precision medicine (Uhlén et al, 2016). In the HPA project, mapping over 90% of protein-coding genes has already been completed, and a reference framework that continues to evolve with the integration of new datasets has been established (Basha et al, 2017; Bertolini et al, 2023; Yu et al, 2015) (Table 1).

The Human Cell Atlas (HCA), launched in 2016, aims to create a detailed reference map of all human cells, the building blocks of life, and advance our understanding of cellular diversity, organisation, and interactions by characterising cell types through their genomics, epigenomics, transcriptomics, and proteomics profiles (Regev et al, 2017). The HCA reveals how cells function within tissues and adapt during development, ageing and disease progression by utilising cutting-edge technologies like single-cell RNA sequencing, spatial transcriptomics and advanced imaging (Heimberg et al, 2024; Itai et al, 2024). With ongoing efforts to expand its content, the freely accessible HCA provide insights into human health and disease (Amit et al, 2024) (Table 1).

Characterising the phenotype-based only on genomics or other single omics is challenging. To accelerate this process, it is crucial to develop a multi-omics approach by generating genomics, epigenomics, transcriptomics, proteomics, metabolomics, lipidomics, metagenomics and phenomics data and integrating these omics data using systems biology and AI. In this context, several pioneering studies (Piening et al, 2018; Price et al, 2017; Tebani et al, 2020) have demonstrated that generating multi-omics data is beneficial for detailed patient characterisation. In the P100 Wellness study, multi-omics data were generated for 108 subjects to follow up with the participants over a 9-month period (Price et al, 2017). In the iPOP study, a multi-omics analysis was performed on 23 carefully selected healthy subjects to describe the systematic changes in weight gain and loss over three visits within a 1-year period (Piening et al, 2018). In both studies, anthropometric, clinical chemistry, plasma proteomics, plasma metabolomics and gut metagenomics data were generated, and an integrative analysis of the data was performed. It is also critical to probe the stability of molecular profiles among healthy individuals over time.

Moreover, a longitudinal wellness cohort comprising 100 healthy Swedish individuals was recruited for 24 months, and analyses were performed for genomics, proteomics, transcriptomics, lipidomics, metabolomics and gut metagenomics (Tebani et al, 2020). The data were also complemented with immune cell profiling and routine clinical chemistry analysis. Overall, the study reported high variation between individuals across different molecular readouts and demonstrated the need for personalised multi-omics analyses for the detailed characterisation of subjects. Hence, the personalised multi-omics data-driven approach may lead to the development of a new medicine that will transition from its current focus on disease to prevention and wellness.

Multi-omics profiling of healthy subjects and patients involves an interdisciplinary group of scientists from different international centres (including universities, hospitals and companies) (Fig. 4). All the biological samples in multi-omics studies should be collected using standardised protocols and kept in biobanks. Multi-omics data should be generated in standardised data generation centres, analysed and integrated using standardised data analysis pipelines, and stored in secure storage centres. This procedure also requires implementing systematic data quality and safety assurance procedures.

### The UK Genome Project

Global WGS projects are being conducted for various ethnic groups worldwide. The world's largest population-based genetics research project in the UK involves >500,000 volunteers (Callaway, 2023). This project compiled deep phenotyping, genomics, and other omics data from all volunteers into an online repository called the UK Biobank (UKBB) (Bycroft et al, 2018; Sudlow et al, 2015). The collection of such big data supports the potential for exploring human variants, especially very rare variants, and their associations with health and disease. Examples include the Mendelian Randomisation study on insomnia symptoms (Gibson et al, 2023) and the regression analysis of colorectal cancer (Bradbury et al, 2020). Additionally, the genomic data in the UKBB database are combined with other omics data, such as the recent association study with the plasma proteome from the UK Biobank Pharma Plasma Proteome (UKB-PPP). Comprehensive protein quantitative trait locus (pQTL) mapping enabled the determination of associations between genetic variants and plasma protein levels and the identification of numerous novel, rare, and ethnicity-specific pQTLs (Dhindsa et al, 2023; Eldjarn et al, 2023; Sun et al, 2023). As a result, the findings of these studies extend the

**Table 1.  Multi-omics data resources for precision medicine approaches.**

| Project | Launch | Conditions | Omics data | Link | References |
|---------|--------|-----------|-----------|------|-----------|
| The FANTOM Consortium | 2000 | Healthy | T | https://fantom.gsc.riken.jp | (Forrest et al, 2014) |
| The Human Protein Atlas | 2003 | Healthy & Cancer | T, SC-T, P | https://www.proteinatlas.org | (Uhlén et al, 2015) |
| ENCODE | 2003 | Healthy & Cancer | EpiG, T | https://www.encodeproject.org | (Abascal et al, 2020) |
| COSMIC | 2004 | Cancer | WGS, EpiG, T | https://cancer.sanger.ac.uk | (Tate et al, 2018) |
| ADNI | 2004 | AD | WGS, Imaging | https://adni.loni.usc.edu | (Petersen et al, 2010) |
| TCGA | 2006 | Cancer | WGS, EpiG, T | https://portal.gdc.cancer.gov | (Weinstein et al, 2013) |
| Roadmap Epigenomics | 2007 | Healthy | EpiG, T | http://www.roadmapepigenomics.org | (Bernstein et al, 2010) |
| 1000 Genomes Project | 2007 | Healthy | WGS | https://www.internationalgenome.org | (Fairley et al, 2019) |
| UK Biobank | 2007 | Various | WGS | https://www.ukbiobank.ac.uk | (Sudlow et al, 2015) |
| GTEx and eGTEx | 2010 | Healthy | WGS, T | https://www.gtexportal.org | (Consortium et al, 2020) and (Stranger et al, 2017) |
| CPTAC | 2011 | Cancer | WGS, EpiG, T, P | https://cptac-data-portal.georgetown.edu | (Li et al, 2023) |
| CommonMind | 2012 | Schizophrenia, bipolar disorder | WGS, EpiG, T | https://www.nimhgenetics.org/resources/commonmind | (Hoffman et al, 2019) |
| PsychENCODE | 2015 | Neuropsychiatric disease | WGS, T | https://psychencode.synapse.org | (Jourdon et al, 2021) |
| MoTrPAC | 2016 | Healthy | WGS, T, P, M, L | https://www.motrpac.org | (Sanford et al, 2020) |
| TARGET | 2016 | Paediatric cancer | WGS, EpiG, T | https://ocg.cancer.gov | (Grossman et al, 2016) |
| The Human Cell Atlas | 2016 | Various | EpiG, T, SC-T, P | https://www.humancellatlas.org/ | (Regev et al, 2017) |
| OncoKB | 2016 | Cancer | WGS | https://www.oncokb.org/ | (Suehnholz et al, 2024) |
| eQTLGen consortium | 2018 | Various | WGS, T | https://www.eqtlgen.org | (Võsa et al, 2021) |
| All of Us | 2018 | Healthy | Surveys, EHRs, Wearables | https://www.researchallofus.org | (Bick et al, 2024) |
| AMP-PD | 2018 | AD, PD, T2D, rheumatoid arthritis, systemic lupus erythematosus | WGS, EpiG, T, P, M | https://amp-pd.org/about | |
| The 10 K Study: a large-scale prospective longitudinal study | 2020 | Healthy | WGS, T, P, M, L, O-MG, G-MG. | | (Shilo et al, 2021) |
| The Human Phenome Initiative in the USA | 2023 | Various | WGS, T, P, M, L, G-MG. | | (Yurkovich et al, 2023) |
| The Anatolian Precision Medicine Initiative (APMI) | 2023 | Various | WGS, T, P, M, L, O-MG & G-MG | https://mphenome.org/apmi | |
| The Global 1 Million Phenome Initiative (1M-PI) | 2025 | Various | WGS, T, P, M, L, O-MG & G-MG | https://mphenome.org/ | |

*WGS* whole-genome sequencing, *EpiG* epigenomics, *T* transcriptomics, *SC-T* single-cell transcriptomics, *P* proteomics, *M* metabolomics, *L* lipidomics, *O-MG* oral metagenomics and *G-MG* gut metagenomics, *AD* Alzheimer's disease, *PD* Parkinson's disease, *T2D* type 2 diabetes.

knowledge about the genetic proxies of several pivotal protein targets for drug discovery, such as PCSK9. These association studies provide new insights into genetic perturbations affecting proteomic pathways, which may facilitate the discovery of potential therapeutic targets (Li, 2023).

In addition to the UKBB study, other WGS projects, such as the UK100,000 Genome Project, are also being developed to investigate the role of WGS in diagnosing rare diseases, for which diagnoses were previously unavailable. As the pilot project, involving 660 participants, proceeded, ~14% of new diagnoses were assigned to previously undiagnosed patients because the relevant pathogenic mutations were poorly covered by other sequencing methods (Smedley et al, 2021). Additionally, sequencing solid tumours from children has demonstrated the clinical utility of integrating WGS

into routine NHS testing for paediatric cancer (Trotman et al, 2022). In summary, patients with diverse diseases can receive focused clinical care with the help of these new diagnostic results from a multi-omics-based approach.

### The Human Phenome Initiative

The Human Phenome Initiative (HPI) in the United States, launched by Phenome Health, a nonprofit research organisation, aims to develop innovative science and bring a focus to wellness, prevention, and healthy ageing (Yurkovich et al, 2023). The data-driven approach performed in the HPI aims to harness new technologies to understand the complex biology of health and different disease states. The HPI plans to conduct longitudinal multi-omics and detailed phenomics analyses for one million

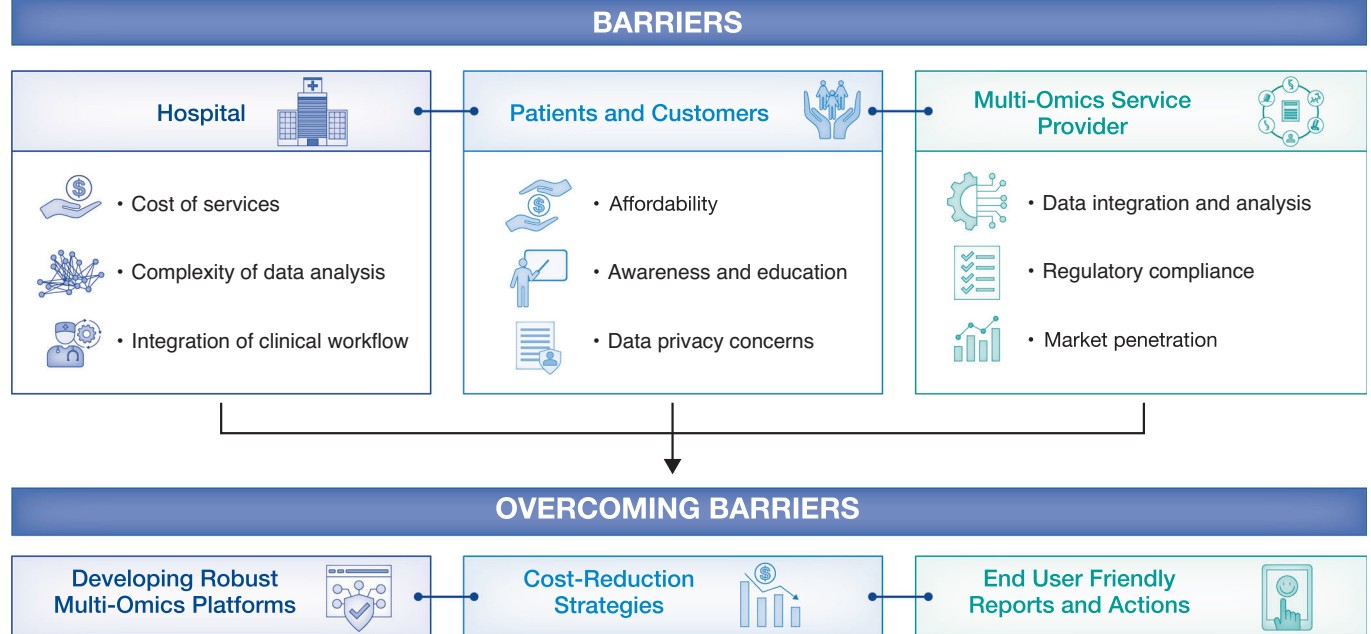

**Figure 4. Overcoming barriers to clinical application of multi-omics data.**

As multi-omics technologies advance, several barriers must be addressed to harness their potential in healthcare. Key challenges include high costs, which limit accessibility, and the complexity of analysing vast and heterogeneous multi-omics data. Technological advancements and economies of scale are necessary to overcome these challenges, enabling cost reduction, while user-friendly analytical tools are crucial for practical data interpretation. Integrating multi-omics data into clinical workflows is challenging due to issues with interoperability, clinician training, and regulatory compliance, all of which are critical for the widespread clinical use of this data. Additionally, increasing awareness and education, along with robust data privacy measures, are vital for ensuring the responsible use of sensitive biological data. Overcoming these barriers through collaboration and innovation will be crucial for expanding market penetration and enabling the full potential of multi-omics in precision medicine. This will ultimately facilitate the successful integration of multi-omics into mainstream healthcare for personalised, data-driven approaches.

individuals over a ten-year period. The HPI ecosystem, which includes the Buck Institute for Research on Aging, will perform longitudinal studies, such as analyses of the plasma proteome, metabolome and lipidome, gut microbiome composition and function, and lifestyle, behaviour, and social determinants of health through wearable devices, self-reported questionnaires and EHRs. By assessing and optimising the health trajectory of each individual using a data-driven personalised approach, the HPI aims to address current challenges in medicine. With this initiative, Phenome Health and its partner ecosystem aim to enhance the quality of healthcare while reducing the cost for healthcare systems by demonstrating the potential of longitudinal big biological data.

**The 10K study: a large-scale prospective longitudinal study**

Healthcare management faces several challenges, including an ageing population, an increased incidence of chronic diseases, rising costs, and marginal quality metrics. The 10K large-scale prospective longitudinal study, established in Israel, aims to develop models of disease onset and progression and to discover novel drug targets and molecular markers to develop effective treatment strategies (Shilo et al, 2021). About 10,000 healthy subjects were recruited, and follow-up visits were scheduled annually for 25 years. The 10K cohort includes a deeply phenotyped longitudinal cohort of healthy individuals aged between 40 and 70. Patients with predefined medical conditions were excluded from the study. At baseline, medical history, lifestyle and nutritional habits, vital signs, anthropometrics, blood test

results, and detailed imaging data were collected. The 10K team performed transcriptomics, proteomics, metabolomics, gut and oral metagenomics, and immunomic analyses for detailed molecular profiling. Glucose levels were also continuously measured using a continuous glucose monitoring device for 2 weeks, and a home sleep apnoea test device monitored sleep for three nights (Keshet et al, 2023). The 10K team aims to transform current medical practice from a disease-oriented to a data-driven, wellness-oriented and personalised population health approach.

BA derived from molecular and physiological biomarkers is a more accurate measure of ageing since ageing varies among individuals of the same chronological age. The 10K-based study developed BA scores across various physiological systems, uncovering sex-specific ageing patterns using machine learning (Reicher et al, 2024). The authors reported that higher BA scores correlated with an increased prevalence of age-related conditions, underscoring their clinical relevance. The study also highlighted system-specific ageing dynamics and the potential of comprehensive cohorts to advance the understanding of chronic diseases, paving the way for personalised prevention strategies.

**The All of Us**

The All of Us Research Program is a large-scale initiative to enrol more than 1 million diverse participants in the United States to accelerate biomedical research and improve health (2019). The program was launched in 2018 and seeks to address gaps in observational cohort studies by recruiting participants from

underrepresented groups and gathering comprehensive data, including health questionnaires, EHRs, biospecimens and data from digital health technology. The program prioritises diversity, with over 80% of participants from underrepresented populations. As of 2024, it had enroled more than 850,000 participants and collected EHRs from more than 550,000 individuals.

The program aims to create a data repository that reflects individual differences in lifestyle, environment, socioeconomic factors, and biology, thereby advancing precision medicine. This data will support research into disease prevention, diagnosis, and treatment. The All of Us program does not focus on specific diseases but includes various health conditions to enable comprehensive studies across different demographic and social contexts. The program prioritises diversity in its recruitment, considering factors like race, ethnicity, age, sex, gender identity and access to healthcare. It also involves participants in shaping the program and ensures that underrepresented populations are targeted for biospecimen collection and physical measurements. Additionally, it collects data through mobile health devices, sensors, and surveys, making the findings accessible to approved researchers for long-term studies to improve health outcomes.

### The Anatolian precision medicine initiative

In the first phase of the Anatolian precision medicine initiative (APMI) performed in Turkiye, the consortium will develop examples for implementing personalised medicine in the healthcare system and ensure that preventive and predictive healthcare models are applicable in clinical practice. The development of targeted diagnosis, follow-up and treatment strategies, identification of risk factors and prevention of chronic and acute diseases may enable the more efficient use of financial resources. These strategies will enhance the quality of clinical practice, quality of life and life expectancy for individuals. In the APMI, a wide range of phenotypic differences will be explored. The APMI will include a large population of healthy individuals and patients diagnosed with rare, metabolic, neurodegenerative or autoimmune diseases and cancer. In the first phase of the APMI, a total of 10,000 healthy individuals and patients diagnosed with >150 different conditions (cancers and rare and complex diseases) will be recruited, and biological specimens, including blood, urine, saliva and faecal samples, from all subjects involved in the study, will be collected. Tissue samples will also be collected where applicable. Unlike other WGS projects, transcriptomics (tissues and whole blood), proteomics (tissue and plasma), metabolomics (plasma and urine), lipidomics (plasma and urine) and metagenomics (saliva and faeces) data will be generated together with WGS and detailed phenomics (clinical and dietary) data. These datasets will be integrated using systems biology and AI to characterise individual patients and diseases in detail. These data will also be combined with the clinical and medical history of the patients. As of December 2024, more than 8,000 subjects have been recruited. The first phase of the APMI is expected to be completed by the end of 2025, and the second phase, which involves recruiting more than 100,000 healthy individuals and patients in Türkiye, will be initiated as part of the Global 1 Million Phenome Initiative.

### The Global 1 Million Phenome Initiative

The Global 1 Million Phenome Initiative (1M-PI) has been launched to transform healthcare by leveraging big biological data,

systems biology, and AI. In the 1M-PI, one million healthy individuals and patients diagnosed with various diseases will be recruited from the USA, Turkiye, South Korea, India and other participating countries. Biological specimens, including blood, urine, saliva and faecal samples, will be collected from all subjects involved in the study. Phenomics, WGS, transcriptomics, proteomics, metabolomics, lipidomics and metagenomics data will be generated following the sample collection and data generation protocols used in the APMI project.

The 1M-PI aims to apply its outcome to clinical practice. The collection of biological specimens, data generation, and analysis were standardised in all participation centres, allowing the data generated in the 1M-PI to be used to design guided treatments and enhance clinical practice globally. In this context, the 1M-PI will collaborate with several pharmaceutical and biotech companies that have extensive experience in the integrative analysis of multi-omics data, generating patient reports based on multi-omics data and utilising this big biological data in AI-based diagnosis and drug development. Thus, the 1M-PI aims to reduce or eliminate the socioeconomic burden associated with these diseases. The 1M-PI team anticipates that their initiative may be a prime example of applying multi-omics profiling in the clinic and advancing clinical practice by establishing precision medicine. Taken together, the path to personalised health may be paved with big biological data, systems biology and AI.

## Creating digital twins using big biological data

Digital twins have emerged as transformative tools in advancing personalised medicine. They offer a novel framework for integrating big biological data into patient-specific models. By computationally creating high-resolution replicas of patients, digital twins can simulate the effects of thousands of therapeutic options, enabling the selection of optimal treatments tailored to individual genetic and physiological profiles (Björnsson et al, 2019). This approach bridges the gap between the complexity of multi-omics data and clinical decision-making, promising significant strides in extending health spans and managing diseases (Jiang et al, 2018; Shalek and Benson, 2017) (Fig. 3).

Central to the utility of digital twins is their capacity to integrate and analyse data from multiple scales and sources, such as metabologenomics, wearable devices, and EHRs (Gawel et al, 2019). These frameworks allow for the dynamic modelling of disease mechanisms across cellular, tissue and systemic levels. By identifying key molecular drivers through network tools, such as GEMs, PPINs, GRNs, SNs and CNs, digital twins enable the prioritisation of therapeutic targets (Tao and Qi, 2019). This methodology is particularly valuable for longevity research, as it uncovers interventions that mitigate cellular ageing and systemic decline, paving the way for personalised strategies that enhance resilience and prolong health spans (Smillie et al, 2019).

Scenario-based modelling within digital twins extends their application to predicting and preventing age-related conditions. By incorporating longitudinal data, digital twins can simulate the progression of chronic diseases, evaluating how lifestyle interventions, environmental factors and pharmaceutical treatments influence outcomes over time (Zhou et al, 2014). This predictive capability informs early interventions to prevent irreversible damage and facilitates the development of therapies targeting

hallmark ageing processes, such as inflammation, cellular senescence, and metabolic dysregulation (Shalek and Benson, 2017). Digital twins hold transformative potential for personalising longevity strategies in precision medicine through these integrative and forward-looking approaches (Coorey et al, 2021; Hernandez-Boussard et al, 2021).

To move beyond summarising existing applications, we propose a forward-looking analytical framework emphasising the convergence of multi-scale modelling, longitudinal monitoring, and individualised prediction. This framework leverages digital twins, also defined as computational replicas of individuals that evolve with time, as dynamic vehicles to integrate systems biology models (e.g. GEMs and other biological networks), real-time data streams (e.g. wearable devices, EHRs), and AI-driven inference. This approach enables mechanistic insight, prospective prediction and adaptive intervention by embedding causal modelling and continuous feedback loops. Furthermore, integrating explainable AI methods within this framework promotes interpretability and trust, which are critical for clinical adoption. This holistic, person-centric paradigm represents a shift from static snapshot analyses to dynamic, context-aware systems supporting proactive, precision healthcare and precision medicine.

## Drug development using big biological data

The traditional drug discovery pipeline is a resource-intensive and time-consuming process, often requiring up to 15 years and an average investment of $1 billion to bring a single drug from initial discovery to regulatory approval. Drug development adheres to a rigorous, multi-phase framework that encompasses target identification and validation, lead compound discovery, preclinical evaluation of pharmacodynamics, pharmacokinetics, efficacy and toxicity, followed by sequential phases of clinical trials in human subjects. With technological advancements, numerous successful multi-omics-based strategies have been implemented at each stage of the drug development process to develop the most effective drug candidates for preclinical studies (Fatima et al, 2024). Although it is a significant achievement to advance a drug candidate to a Phase I clinical trial after substantial work at the preclinical stage, 90% of drug candidates fail during clinical trials and the drug approval process. Analyses of clinical trial data suggest that a lack of clinical efficacy, toxicity, and poor drug-like properties may be potential reasons for clinical failure in drug development. Hence, multi-omics-based strategies can be implemented to optimise and validate the drug development process, thereby increasing the overall success rate of drug development (Vamathevan et al, 2019).

Most target discovery and validation studies have been performed using animal models. However, animal models often cannot recapitulate the complete mechanism of human diseases, leading to translational errors at the clinical stage of drug development due to their low efficacy. There are also challenges related to the heterogeneity of the patient population, which might be alleviated with detailed clinical phenotyping using multi-omics data. The unknown pathophysiology of a wide range of complex diseases, including cancer, metabolic, neurodegenerative and autoimmune diseases, also makes reliable target identification challenging due to the genetic, behavioural and environmental factors involved in the progression of these diseases. In this context, the synergy between the big biological data of complex diseases,

systems biology, and AI has the potential to accelerate drug development by providing insights into disease biology, facilitating the identification and validation of reliable drug targets, expediting the discovery of novel compounds and enabling the development of more targeted and personalised therapies for various diseases. Integrating big biological data, systems biology and AI in patient characterisation within clinical trials may also enhance the selection of appropriate patients, improve treatment efficacy assessments, and contribute to personalised medicine by identifying tailored treatments for specific patient groups, ultimately advancing clinical trial efficiency and success rates.

Although the drug discovery programmes integrate the different stages of the research and development process, we split the entire programme into four distinct elements: targets, drugs, animals and patients. As discussed above, big biological data have been successfully used to discover novel and reliable targets in combination with systems biology and AI. After assessing the limitations of clinical trials, similar strategies may be implemented immediately. We envision that we can overcome specific obstacles during the successful completion of Phase III clinical trials by generating big biological data during Phase I and II clinical trials. Generating big biological data from early-stage clinical trials may reveal these drugs' positive and side effects on patients and characterise patient responses to treatment. Moreover, such big biological data may also help to identify novel biomarkers for stratifying patients who respond and those who do not.

Despite the significant effort and funding, developing effective drugs for many complex diseases remains a challenging task. Hence, generating big biological data and using this knowledge with systems biology and AI may lead to success in drug development.

## Diagnosis using big biological data

Integrating big biological data, systems biology and AI may enable the development of more precise diagnostic tools and tailored treatment plans for improved health outcomes. Combining these methods may also facilitate personalised diagnostic approaches by considering an individual's unique genetic, molecular, behavioural and clinical profiles for better disease management and treatment selection. Such integration can enhance the early detection of diseases, enable stratification of patient populations at risk of or susceptible to certain diseases, significantly improve accuracy, and facilitate personalised preventive care and treatment, revolutionising how diseases are diagnosed and managed in healthcare. Diagnosis/screening using big biological data may generate actionable results in the near term compared to drug development.

Comprehensive genomic profiling has been widely adopted to identify genetic alterations, enabling precise diagnosis, prognosis, and targeted treatment strategies across various diseases, particularly in oncology (Chakravarty and Solit, 2021; George et al, 2015). Newborn screening involves a range of tests and assessments performed shortly after birth to evaluate the health, development, and potential risks or conditions affecting the newborn. Early identification allows healthcare providers to promptly initiate treatments or interventions, potentially minimising the impact of specific conditions on a child's health and well-being. The technologies used for newborn screening have undergone exponential development in recent decades. The first developed

newborn blood test was for phenylketonuria, while the application of tandem mass spectrometry enabled testing for ~50 inborn errors of metabolic disorders (Stark and Scott, 2023). WGS-based expanded newborn screening is widely used in the clinic, providing risk information for thousands of early-onset rare diseases. WGS has been implemented nationwide, with multiple cases of successful early diagnosis that have helped affected newborns receive timely treatment (Kingsmore et al, 2022; Owen et al, 2022; Seydel, 2022). However, the interpretation of WGS data also brings additional challenges. Newborn pilot trials, such as the BabySeq project (Ceyhan-Birsoy et al, 2019), identify many variants with uncertain pathogenicity and low prevalence.

Olin et al (2018) performed longitudinal analyses of 100 newborns by sampling up to four times during their first 3 months of life. The authors analysed 58 immune cell populations by mass cytometry and 267 plasma proteins by immunoassay. They reported that preterm and term children differ at birth but converge on a shared trajectory, driven by microbial interactions and hampered by early gut bacterial dysbiosis. This requires further multi-omics analysis of newborns to determine their health.

Most rare diseases can be identified and potentially eradicated by performing carrier screening on the candidate parents. Carrier screening in a medical context typically refers to identifying and evaluating individuals who carry a genetic mutation for a particular disease or condition, even if they do not exhibit any symptoms themselves. Such identification is often relevant in recessive and X-linked genetic disorders where a person, though not affected by the disease, can pass the mutated gene to their offspring. More than 1000 of these disorders affect an estimated 1 in 300 pregnancies (Hogan et al, 2018; Johansen Taber et al, 2019). Carrier screening involves a family history assessment, genetic testing, and counselling with clinicians and geneticists. The execution of expanded carrier screening using WGS data and analysis has a prominent impact on a couple's reproductive decision-making and enables interventions to prevent affected pregnancies (Johansen Taber et al, 2019). Moreover, there is also a need to generate plasma proteomics and metabolomics data and metagenomics for oral, gut, and vaginal metagenomics data to facilitate a detailed characterisation of the candidate parents and prevent any complications for their offspring.

## Limitations and risks of AI in precision medicine

Integrating heterogeneous big biological data consisting of multi-omics, clinical, wearable device, imaging and environmental data presents substantial practical challenges (Fig. 4). Technical variability can introduce batch effects, especially when data are collected across different platforms, sites, or time points. Additionally, temporal misalignment arising from inconsistent or asynchronous sampling can complicate longitudinal analyses and obscure dynamic biological patterns. Data sparsity, particularly in high-dimensional omics datasets or underrepresented populations, further limits analytical power and generalisability (Chustecki, 2024). Standardised biological sample collection and data generation, analysis, integration and interpretation protocols are critical to address these issues. In this context, establishing standardised multi-omics data generation laboratories worldwide and developing pre- and post-processing pipelines for big biological data analysis are essential. Advanced imputation techniques, including deep learning-based methods, have shown promise in recovering missing

values while preserving biological signals. Moreover, federated learning frameworks allow collaborative model training across decentralised datasets while maintaining data privacy and compliance with local regulations. By incorporating such strategies, researchers can enhance the quality, interpretability and reproducibility of integrated analyses, thereby strengthening the translational impact of precision medicine initiatives (Johnson et al, 2021).

While AI holds immense promise for advancing big biological data interpretation and precision medicine, it is essential to acknowledge its current limitations. Algorithmic bias remains a significant concern, especially when models are trained on datasets that underrepresent specific populations, resulting in disparities in performance and potential inequities in clinical outcomes (Carini and Seyhan, 2024). Data heterogeneity, stemming from differences in experimental platforms, cohort characteristics and data quality, can compromise model robustness and generalisability. Moreover, the reproducibility of AI-generated predictions is often hindered by opaque model architectures, a lack of standardised benchmarking, and limited access to code and training data. These challenges underscore the need for greater transparency, developing explainable AI methods, and establishing rigorous validation frameworks. By addressing these limitations, the field can ensure that AI applications in biology and medicine are robust, equitable, reliable and clinically actionable (Kelly et al, 2019).

While AI offers powerful tools for data-driven healthcare, the risk of misinterpreting AI-generated outputs, especially in high-stakes clinical decision-making, must be carefully considered. Overreliance on AI predictions without appropriate clinical validation or understanding of model limitations can lead to diagnostic errors, inappropriate treatment choices or missed adverse events. The complexity and opacity of many AI models, often referred to as 'black boxes,' exacerbate this risk by limiting interpretability and undermining clinician trust. Therefore, integrating explainable AI techniques and fostering collaboration between AI developers and healthcare professionals is crucial to ensure that AI tools complement, rather than replace, clinical judgement. Rigorous prospective validation, continuous monitoring, and clear communication of uncertainty are necessary to safeguard patient safety and optimise the clinical utility of AI-driven insights.

# Conclusions and future perspectives

Current medical practice prioritises disease treatment over prevention or health optimisation, resulting in substantial costs associated with chronic and late-stage conditions. Longitudinal big biological data, applied from the bench to the bedside, offers transformative potential by advancing our understanding of the biological processes and molecular mechanisms underlying disease progression. This facilitates the development of precision medicine, promising improved diagnostics, therapeutics and longevity.

Despite significant progress, scaling these approaches for widespread clinical application remains challenging. Generating, storing and analysing big biological data requires robust infrastructure and interdisciplinary collaboration. Standardised protocols and validation methods are essential to ensure the reliability and reproducibility of big biological data. Ethical considerations, including data privacy and informed consent, are equally crucial for

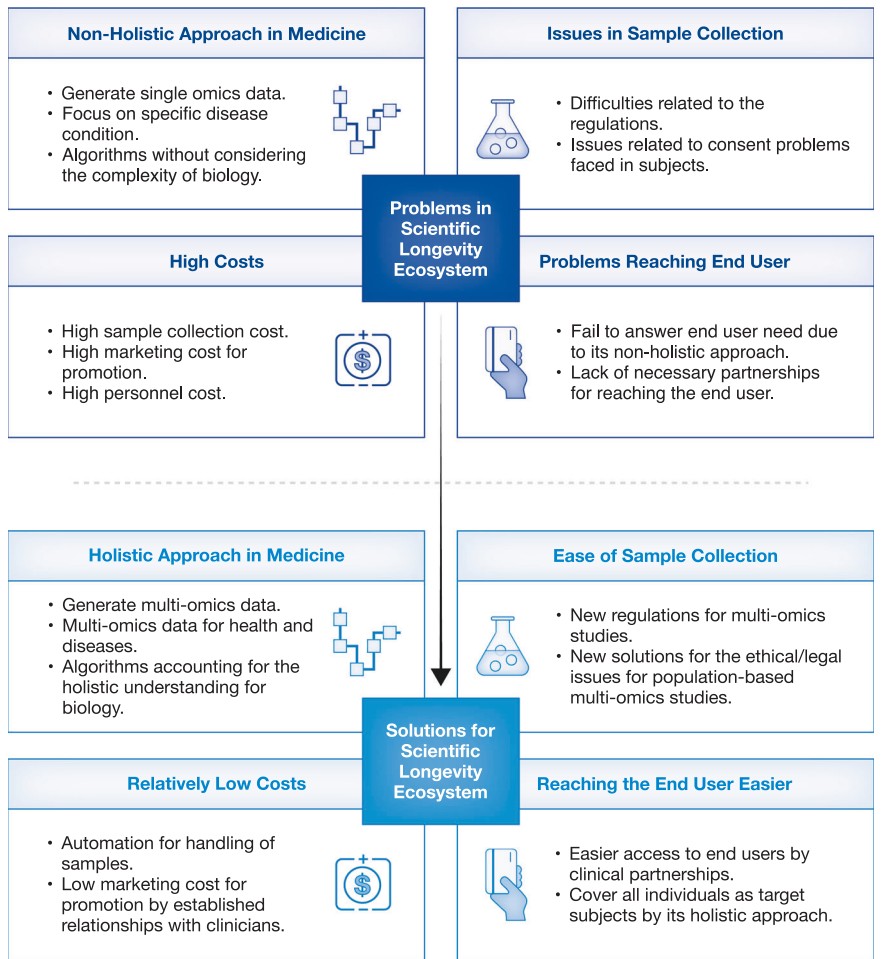

**Figure 5. The solution for running cross-population-based longevity studies.**

The scientific longevity ecosystem faces several challenges, including the non-holistic approach of traditional medicine, which often focuses on treating symptoms rather than addressing root causes and individual factors. High costs associated with sample collection and multi-omics data generation limit the accessibility and scalability. Additionally, sample collection issues, such as procedure variability, can impact data reliability. Recruiting patients for studies is also challenging due to reluctance, ethical concerns and logistical barriers, which can affect the diversity and generalisability of the findings. Overcoming these challenges through technological advancements, cost reductions and improved study designs is crucial for advancing scientific longevity.

the successful implementation of studies in both preclinical and clinical settings. Overcoming these challenges is pivotal to transition from a one-size-fits-all model to a precision health paradigm focused on prevention and wellness (Fig. 5).

Longitudinal big biological data initiatives aim to refine clinical practices and tailor medical treatments to individual needs. Projects such as national genome initiatives, wellness cohorts and multi-omics-based population profiling create valuable frameworks for generating and integrating multi-omics data, bridging the gap between research and clinical practice. These integrative analyses elucidate the molecular basis of diseases, identify genetic and molecular variations and characterise patient phenotypes, enabling more effective medication adjustments and inspiring novel drug discovery. Systematic big biological data analyses also shift the focus from correlation to causation, establishing the foundation for preventive medicine. Success in these approaches relies on cross-disciplinary collaboration, technological innovation and adherence to ethical and regulatory standards.

Multi-omics analysis of parents before pregnancy and newborns may be critical to a 'healthy start in life'. The first year of life is a crucial period in the development of babies, during which they experience rapid and complex changes in various physiological systems. The integrative analysis of longitudinal big biological data from trios (father-mother-child) and environmental data may help identify the molecular components of developmental variations in the first 12 months after birth. In this context, multi-omics data from trios can also be generated in the first year after birth. This large dataset, together with comprehensive phenomic data for each trio, can be analysed multidimensionally to elucidate the critical molecular pathways and networks involved in the development of newborns. Ultimately, this approach may lead to the development of more precise diagnostic and therapeutic tools for each child. Although collecting samples and generating omics data for such a comprehensive study is challenging, developing new analysis methods through systems biology and AI to dissect the molecular components of developmental variations in multiple systems, such

as the cognitive and immune systems, is feasible. An additional, crucial part of such an initiative may be the collection of all relevant exposome data, such as maternal health conditions, drug use and socioeconomic status, for each family involved in the study. The identified molecular components of these developmental variations may eventually create unprecedented opportunities to ameliorate childhood problems and provide better chances of living healthier lives. This approach may be the most assertive and comprehensive attempt at understanding the molecular basis of human development in the first year after birth.

GEMs are crucial in elucidating the underlying molecular mechanisms of diseases, identifying drug targets and biomarkers and tailoring patient-specific treatments. Combining GEMs with other biological networks, which are built on big biological data, supports the creation of digital twins and provides a comprehensive framework for integrating diverse datasets, fostering a holistic understanding of biological systems and their dysfunctions in disease states. They also enable insights into disease progression, facilitating the creation of advanced diagnostic tools and therapeutic strategies. End-to-end computational platforms for analysing complex datasets are crucial for characterising health and disease states, elucidating molecular mechanisms, and discovering novel biomarkers and therapeutic targets.

Integrating big biological data, systems biology and AI can revolutionise our understanding of complex diseases, paving the way for more personalised and effective healthcare solutions. This multi-dimensional approach deciphers the molecular mechanisms underlying disease progression, identifies novel drug targets and biomarkers and develops targeted and effective therapeutic strategies. AI-driven models analyse and integrate multi-omics data with other big biological data sources, such as EHRs, wearable devices and environmental factors, including diet, drugs and toxins. These tools offer unprecedented insights into cellular metabolism and host–microbiome interactions, significantly enhancing our grasp of metabolic pathways and other biological functions. Moreover, AI's ability to mine biomedical literature and integrate longitudinal big biological data enhances our capacity to develop precision medicine approaches tailored to individual patients. This integration enables the development of predictive models that can model disease progression, treatment responses and personal health risks, thereby improving diagnostic accuracy and timeliness.

Precision medicine aims to customise interventions for rare and complex diseases based on individual characteristics, including multi-omics, lifestyle and environment (Torkamani et al, 2017). The synergy between big biological data, systems biology and AI is pivotal in advancing precision medicine, offering more profound insights into individual health profiles and enabling personalised therapeutic approaches. Big biological data integration also uncovers the aetiology of complex diseases through genome-first and phenome-first approaches, providing a comprehensive understanding of pathological anomalies and molecular underpinnings (Aron-Wisnewsky et al, 2020).

As AI-driven tools become increasingly integrated into healthcare, addressing regulatory, privacy and ethical considerations is essential for responsible implementation. Regulatory frameworks, such as those from the US Food and Drug Administration (FDA) and the European Medicines Agency (EMA), are beginning to adapt to the evaluation of AI-based clinical decision support systems, including requirements for transparency, robustness and post-deployment monitoring (Palaniappan et al, 2024). Ensuring compliance with data privacy regulations like HIPAA and GDPR is especially critical when handling sensitive longitudinal big biological data. Using digital twins of individuals that continuously integrate multi-modal data raises novel ethical questions regarding informed consent, data ownership and the potential for algorithmic determinism in health predictions (Xu et al, 2024). These concerns underscore the need for robust ethical frameworks, participatory design that involves patients and clinicians, and governance structures that prioritise fairness, explainability and trustworthiness in AI systems. Addressing these dimensions may translate technological advances into equitable and clinically viable precision healthcare solutions.

In conclusion, integrating big biological data, systems biology and AI represents a paradigm shift in characterising health and disease states and developing effective treatment strategies. We highlighted the emerging big biological data modalities for precision health and discussed their clinical applications. The application areas outlined above demonstrate that we are just beginning to realise the value of applying the combination of big biological data, systems biology and AI to critical healthcare problems. As omics technologies, data analytics and computational tools continue to evolve, they promise to broaden the horizon of multi-omics research. By fostering interdisciplinary collaborations and leveraging cutting-edge technologies, this approach promises to revolutionise and transform healthcare. It enables a holistic understanding of disease, personalised treatment modalities and enhanced patient care, heralding a new era of precision medicine focused on prevention, wellness and individualised care. As these fields evolve, they promise to revolutionise 21st-century medicine by offering enhanced patient care, individualised treatment, and a new era of systems biology and AI-based healthcare models focused on prevention and wellness (Auton et al, 2015; Freimer and Sabatti 2003; Omiye et al, 2024).

## Peer review information

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

## Acknowledgements

AM and MU acknowledge the support from the Knut and Alice Wallenberg Foundation under grant number 72254. The National Research Foundation of Korea supported MS under grant number NRF-2023-R1A2C3003438. The authors thank Meng Yuan and members of the Systems Medicine group for their comments about the manuscript.

## Author contributions

**Adil Mardinoglu**: Conceptualisation; Writing—original draft; Writing—review and editing. **Hasan Turkez**: Investigation; Writing—review and editing. **Minho Shong**: Conceptualisation; Investigation; Writing—review and editing. **Vishnuvardhan Pogunulu Srinivasulu**: Conceptualisation; Investigation; Writing—review and editing. **Jens Nielsen**: Conceptualisation; Investigation; Writing—review and editing. **Bernhard O Palsson**: Conceptualisation; Investigation; Writing—review and editing. **Leroy Hood**: Conceptualisation; Investigation; Writing—review and editing. **Mathias Uhlen**: Conceptualisation; Investigation; Writing—review and editing.

## Funding

## Disclosure and competing interests statement

AM co-founded SZA Longevity, Trustlife Therapeutics, ScandiBio Therapeutics and ScandiEdge Therapeutics; MS co-founded SILK Longevity; VPS co-founded Vizzhy Longevity; LH co-founded Phenome Health; BOP co-founded Pasteur 21, Sinopia Biosciences and Conarium Bioworks; and MU co-founded ScandiBio Therapeutics, ScandiEdge Therapeutics and ProteomEdge.

