## [Peer Review File · Molecular Systems Biology]

Longitudinal big biological data in the AI era

Adil Mardinoglu, Hasan Turkez, Minho Shong, Vishnuvardhan Srinivasulu, Jens Nielsen, Bernhard Palsson, Leroy Hood, and Mathias Uhlen

Corresponding author(s): Adil Mardinoglu (adilm@scilifelab.se)

Review Timeline:

Submission Date:	21st Mar 25
Editorial Decision:	14th May 25
Revision Received:	7th Jun 25
Editorial Decision:	30th Jun 25
Revision Received:	30th Jun 25
Accepted:	2nd Jul 25

Editor: Jingyi Hou

Transaction Report:

14th May 2025

Manuscript Number: MSB-2025-12981-T
Title: Longitudinal big biological data in the AI era
Author: Adil Mardinoglu
Hasan Turkez
Minho Shong
Vishnuvardhan Srinivasulu
Jens Nielsen
Leroy Hood
Bernhard Palsson
Mathias Uhlen

Dear Adil,

Thank you for submitting your interesting and timely Review article to Molecular Systems Biology. I would like to apologise for the delay in the review process.

So far, we have received reports from two of the four reviewers who agreed to evaluate your manuscript. Despite several reminders, we have not received a response from Reviewer #4. Reviewer #3 has committed to providing feedback, but it has not yet been submitted.

Given the circumstances and the fact that the two available reviews are largely aligned in their recommendations, I have decided to proceed with a decision at this stage to avoid further delays. Should we receive comments from Reviewer #3 at a later point, we will forward them to you, and you can address their points alongside those raised by Reviewers #1 and #2.

Both reviewers find your manuscript relevant and of interest but raise several important points that should be addressed in a revised version. We believe their suggestions are clear and constructive, and we are confident that addressing them will significantly strengthen your article. We therefore strongly encourage you to consider all the feedback carefully in your revision.

Please note that our graphics designer will edit the figures. I will forward the revised manuscript to them once it has been resubmitted, as you may wish to update the figures during the revision process.

If you feel you can satisfactorily deal with these points and those listed by the referees, you may wish to submit a revised version of your manuscript. Please attach a covering letter giving details of the way in which you have handled each of the points raised by the referees. A revised manuscript will be once again subject to review and you probably understand that we can give you no guarantee at this stage that the eventual outcome will be favorable.

We look forward to receiving your revised manuscript.

Kind regards,
Jingyi

Jingyi Hou, PhD
Senior Editor
Molecular Systems Biology

*** PLEASE NOTE *** As part of the EMBO Press transparent editorial process initiative (see our Editorial at <https://dx.doi.org/10.1038/msb.2010.72>), Molecular Systems Biology publishes online a Review Process File with each accepted manuscripts. This file will be published in conjunction with your paper and will include the anonymous referee reports, your point-by-point response and all pertinent correspondence relating to the manuscript. If you do NOT want this File to be published, please inform the editorial office at contact@molsystbiol.org within 14 days upon receipt of the present letter.

Reviewer #1:

This review explores how longitudinal and multi-layered big biological data by encompassing multi-omics, clinical, wearable device-generated, and environmental data can be harnessed through AI and systems biology to enhance our understanding of health and disease. It highlights recent applications in digital twin development, biomarker and drug target discovery, and precision medicine. The review discusses key initiatives, such as the Human Protein Atlas, UK Biobank, the 10K Study, and the Anatolian Precision Medicine Initiative, which aim to integrate such data at scale. Computational tools, including genome-scale metabolic models (GEMs) and machine learning approaches, are emphasized as essential for modelling complex diseases and informing diagnostics and therapies. Ultimately, the paper advocates for a shift from reactive to preventive, personalized healthcare enabled by integrated big data platforms and AI.

The review provides a comprehensive and timely overview of how longitudinal big biological data, AI, and systems biology can be integrated to advance precision medicine. The overall structure is logical, and the review includes several relevant and illustrative examples. Below are a few critical comments for consideration.

Major comments

Page 4: The review briefly mentions large language models (LLMs) in the context of biomedical text mining but does not elaborate on specific models such as GPT or BioBERT, in particular their potential roles in hypothesis generation, clinical decision support, or literature-based discovery. Given the rapid progress in this area, including a discussion on the emerging applications of LLMs along with relevant studies within the scope of the review would be valuable. For instance, in the section "AI for the interpretation of multi-omics data," expanding on their applications (e.g., GPT-based models for biomedical literature mining, clinical report summarization, or integration with omics analysis pipelines if any) could provide additional context and highlight recent advances in this fast-evolving field.

Pages 4 and 5, "Systems biology for the integration of multi-omics data": While GEMs are a central tool in systems biology, the integration of multi-omics data have benefitted from additional approaches such as co-expression and regulatory networks and probabilistic graphical models. Including an additional discussion of these methods would broaden the scope and enhance the utility of the systems biology section.

Page 17: While the review acknowledges the need for infrastructure and standardization, it does not deeply address the practical challenges associated with integrating heterogeneous data types (e.g., batch effects, temporal misalignment, and data sparsity). Including a brief discussion of these limitations, along with potential mitigation strategies (e.g., data harmonization protocols, imputation methods, or federated learning), would strengthen the review and enhance its practical relevance.

Page 31, Figure 1: The figure contains decorative elements that do not clearly convey specific scientific information. Simplifying the design and emphasizing data-driven or conceptually relevant visuals would improve clarity and enhance the figure's scientific value.

Page 31-33, Figures 1-3: removing or simplifying the shading could improve clarity and help the viewer focus on the key conceptual elements.

Figures: All the current figures are high-level, summarizing, and largely symbolic. To enhance the scientific depth and grounding of the review, it would be valuable to include one or two data-driven figures that directly reflect key findings from some of the cited studies that are central to the manuscript's narrative. This would help illustrate the practical application and impact of the reviewed approaches more concretely.

Minor comments

Page 2, Abstract: In the abstract, the word "wearable" does not match the format of the other data types such as multi-omics or clinical data. It might be clearer to say "wearable device data" or "wearable device data" to keep the wording consistent.

Page 8: The section title "Initiatives for personalised population big biological data" sounds a little awkward. Can the authors consider revising it such as "Population-scale initiatives for personalised big biological data"?

Reviewer #2:

The paper provides a comprehensive overview of the integration of artificial intelligence (AI) and systems biology for interpreting multi-omics data in health and disease. It emphasizes the significance of longitudinal big biological data in advancing precision medicine, digital twins, and AI-driven healthcare. The authors discuss major technological advancements in multi-omics, computational modeling, and AI applications, highlighting their transformative potential in diagnostics, drug discovery, and clinical decision support.

It addresses the growing intersection of AI, systems biology, and precision medicine, aligning well with current research priorities and technological advancements. The discussion on digital twins is particularly valuable, as this concept is gaining traction in computational medicine. And it references a broad range of studies, providing substantial background for readers unfamiliar with

the field.

Many of the AI and systems biology applications discussed have been covered extensively in previous literature. A more innovative perspective or unique analytical framework would enhance its impact.

The paper provides an optimistic view of AI applications but does not sufficiently discuss the limitations of AI in big biological data analysis.

Issues such as algorithmic bias, data heterogeneity, and the reproducibility of AI-generated predictions should be addressed. The discussion on GEMs is well-developed, but the paper does not explore alternative computational models that might also contribute to multi-omics data integration.

Including a comparative analysis of different computational approaches (e.g., deep learning models, graph-based AI models) would add value.

Given the paper's emphasis on AI-driven healthcare, a section on regulatory considerations (e.g., FDA approvals for AI tools, data privacy concerns, and ethical implications of digital twins) would be beneficial.

The potential risks of AI misinterpretation, particularly in high-stakes clinical decision-making, should be acknowledged.

This paper is highly relevant for researchers in computational biology, bioinformatics, and AI-driven healthcare. It serves as a useful review for those looking to understand the intersection of multi-omics data and AI applications.

Dear Editor

We want to thank you for considering our paper and sending it out for peer review. We would also like to thank both reviewers for their thoughtful and constructive feedback. We appreciate the recognition of the relevance and timeliness of our review, and we have carefully revised the manuscript to address all comments. Below, we provide a point-by-point response to each comment.

Response to Reviewer Comments

Reviewer #1:

This review explores how longitudinal and multi-layered big biological data by encompassing multi-omics, clinical, wearable device-generated, and environmental data can be harnessed through AI and systems biology to enhance our understanding of health and disease. It highlights recent applications in digital twin development, biomarker and drug target discovery, and precision medicine. The review discusses key initiatives, such as the Human Protein Atlas, UK Biobank, the 10K Study, and the Anatolian Precision Medicine Initiative, which aim to integrate such data at scale. Computational tools, including genome-scale metabolic models (GEMs) and machine learning approaches, are emphasized as essential for modelling complex diseases and informing diagnostics and therapies. Ultimately, the paper advocates for a shift from reactive to preventive, personalized healthcare enabled by integrated big data platforms and AI.

The review provides a comprehensive and timely overview of how longitudinal big biological data, AI, and systems biology can be integrated to advance precision medicine. The overall structure is logical, and the review includes several relevant and illustrative examples. Below are a few critical comments for consideration.

Major comments

Page 4: The review briefly mentions large language models (LLMs) in the context of biomedical text mining but does not elaborate on specific models such as GPT or BioBERT, in particular their potential roles in hypothesis generation, clinical decision support, or literature-based discovery. Given the rapid progress in this area, including a discussion on the emerging applications of LLMs along with relevant studies within the scope of the review would be valuable. For instance, in the section "AI for the interpretation of multi-omics data," expanding on their applications (e.g., GPT-based models for biomedical literature mining, clinical report summarization, or integration with omics analysis pipelines if any) could provide additional context and highlight recent advances in this fast-evolving field.

Response:

Thank you for this insightful suggestion. We have expanded the section "AI for the interpretation of multi-omics data" to include a dedicated paragraph on LLMs, specifically highlighting models such as GPT, BioBERT, and PubMedBERT. We discuss their applications in biomedical literature mining, automated clinical summarisation, hypothesis generation, and integration with omics workflows. Relevant studies demonstrating these capabilities have been cited. We added the text below:

“Large language models (LLMs), such as GPT, Llama, Gemini, BioBERT, and PubMedBERT, have recently emerged as powerful tools in biomedical informatics (Lu *et al*, 2024). These models, trained on large-scale biomedical corpora, can perform various tasks, including literature mining, clinical report summarisation, and question answering, making them particularly useful for hypothesis generation and clinical decision support. For example, GPT-based models have been used to synthesise insights from vast biomedical literature, enabling the identification of novel gene-disease associations and mechanistic hypotheses (Wang *et al*, 2024). Similarly, BioBERT has shown strong performance in extracting biomedical entities and relationships, supporting the development of structured knowledge graphs that can be integrated with multi-omics data (Rehana *et al*, 2024). Recent efforts have also explored the coupling of LLMs with omics analysis pipelines to contextualise molecular findings within the current scientific landscape, aiding in interpreting results and prioritising biomarkers or drug targets (Toufiq *et al*, 2023). As LLMs evolve, their integration into precision medicine workflows is expected to enhance data interpretation, reduce information bottlenecks, and support more informed clinical decision-making.”

Pages 4 and 5, "Systems biology for the integration of multi-omics data": While GEMs are a central tool in systems biology, the integration of multi-omics data have benefitted from additional approaches such as co-expression and regulatory networks and probabilistic graphical models. Including an additional discussion of these methods would broaden the scope and enhance the utility of the systems biology section

Response:

We agree with the reviewer’s comments and have now revised the systems biology section to incorporate a broader range of integrative modelling approaches. In addition to GEMs, we now include discussions on protein-protein interaction networks, signalling networks, gene co-expression networks, transcriptional regulatory networks, and probabilistic graphical models (e.g., Bayesian networks). We added the text below and cited recent studies that employ these approaches in multi-omics data integration.

“While GEMs have been instrumental in interpreting and simulating cellular metabolism across various contexts, systems biology offers a broader suite of methodologies for multi-omics data integration. GRN inference methods, including those based on transcription factor binding and epigenomic data, provide insight into the hierarchical control of gene expression and the impact of regulatory perturbations (Unger Avila *et al*, 2024). PPINs enrich this landscape by capturing the physical interactions among proteins, which are crucial for executing cellular functions and signal transduction (Ramos *et al*, 2024). SNs map the cascades of molecular interactions triggered by extracellular cues, allowing the dissection of dynamic cellular responses and pathway crosstalk (Garrido-Rodriguez *et al*, 2022). CNs, for instance, enable the identification of gene modules with correlated expression patterns, often revealing coordinated biological functions or shared regulatory control (Lee *et al.*, 2017).

While biological networks offer a mechanistic and interpretable framework for simulating metabolic phenotypes, alternative computational models provide complementary strengths in multi-omics data integration. Deep learning models, such as variational autoencoders and multi-modal neural networks, excel at capturing complex, nonlinear patterns across diverse omics layers and have been used effectively for patient stratification, feature extraction, and disease prediction (Ballard *et al*, 2024; Ryu *et al*, 2018). Graph-based AI models, including graph neural networks (GNNs), are particularly suited for integrating biological networks

(e.g., protein-protein interaction, gene regulatory, or signalling networks) with omics data, enabling the modelling of topological dependencies and higher-order relationships (Valous *et al*, 2024). While often less interpretable than constraint-based models, these data-driven approaches offer scalability and flexibility, especially in high-dimensional, heterogeneous datasets. A comparative view that considers the trade-offs between mechanistic insight, predictive accuracy, and interpretability can guide the choice of computational tools for different research and clinical objectives in precision medicine.

Additionally, probabilistic graphical models, including Bayesian networks, capture conditional dependencies among molecular entities, making them especially valuable for modelling causal relationships and integrating heterogeneous omics layers (e.g., transcriptomics with proteomics or metabolomics) (Jiang *et al*, 2025). These complementary approaches enable the construction of interpretable, data-driven models of biological systems and have been successfully applied in disease subtype classification, biomarker prioritisation, and discovery of novel mechanisms. Integrating diverse systems biology frameworks with AI-driven analytics may enhance our ability to decipher complex biological processes and foster a more comprehensive understanding of health and disease states.”

Page 17: While the review acknowledges the need for infrastructure and standardization, it does not deeply address the practical challenges associated with integrating heterogeneous data types (e.g., batch effects, temporal misalignment, and data sparsity). Including a brief discussion of these limitations, along with potential mitigation strategies (e.g., data harmonization protocols, imputation methods, or federated learning), would strengthen the review and enhance its practical relevance.

Response:

We have added a new paragraph in the infrastructure and standardisation section addressing these integration challenges. We describe common issues such as batch effects, data sparsity, and temporal misalignment. We also outline strategies including data harmonisation, imputation methods, and federated learning approaches. Key references have been added to support this discussion.

“Integrating heterogeneous big biological data consisting of multi-omics, clinical, wearable devices, imaging and environmental data presents substantial practical challenges (Figure 4). Technical variability can introduce batch effects, especially when data are collected across different platforms, sites, or time points. Additionally, temporal misalignment arising from inconsistent or asynchronous sampling complicates longitudinal analyses and may obscure dynamic biological patterns. Data sparsity, particularly in high-dimensional omics datasets or underrepresented populations, further limits analytical power and generalizability (Chustecki, 2024). Standardised biological sample collection and data generation, analysis, integration and interpretation protocols are critical to address these issues. In this context, establishing standardised multi-omics data generation laboratories across the globe and developing pre- and post-processing big biological data analysis pipelines are essential. Advanced imputation techniques, including deep learning-based methods, have shown promise in recovering missing values while preserving biological signals. Moreover, federated learning frameworks allow collaborative model training across decentralised datasets while maintaining data privacy and compliance with local regulations. By incorporating such strategies, researchers can enhance the quality, interpretability, and reproducibility of integrated analyses, thereby strengthening the translational impact of precision medicine initiatives (Johnson *et al*, 2021).

While AI holds immense promise for advancing big biological data interpretation and precision medicine, it is essential to acknowledge its current limitations. Algorithmic bias remains a significant concern, especially when models are trained on datasets that underrepresent specific populations, leading to disparities in performance and potential inequities in clinical outcomes (Carini & Seyhan, 2024). Data heterogeneity, stemming from differences in experimental platforms, cohort characteristics, and data quality, can compromise model robustness and generalizability. Moreover, the reproducibility of AI-generated predictions is often hindered by opaque model architectures, a lack of standardised benchmarking, and limited access to code and training data. These challenges underscore the need for greater transparency, developing explainable AI methods, and establishing rigorous validation frameworks. By addressing these limitations, the field can ensure that AI applications in biology and medicine are robust, equitable, reliable, and clinically actionable (Kelly *et al*, 2019).”

Page 31, Figure 1: The figure contains decorative elements that do not clearly convey specific scientific information. Simplifying the design and emphasizing data-driven or conceptually relevant visuals would improve clarity and enhance the figure's scientific value.

Response:

The journal editors will redraw the figures during production. The revised figures will focus on clearly conveying the relationships among data types, computational tools, and clinical applications in precision medicine.

Page 31-33, Figures 1-3: Removing or simplifying the shading could improve clarity and help the viewer focus on the key conceptual elements.

Response:

The journal editors will redraw the figures during production.

Figures: All the current figures are high-level, summarizing, and largely symbolic. To enhance the scientific depth and grounding of the review, it would be valuable to include one or two data-driven figures that directly reflect key findings from some of the cited studies that are central to the manuscript's narrative. This would help illustrate the practical application and impact of the reviewed approaches more concretely.

Response:

The journal editors will redraw all the figures. Hence, we have not revised the figures.

Minor comments

Page 2, Abstract: In the abstract, the word "wearable" does not match the format of the other data types such as multi-omics or clinical data. It might be clearer to say "wearable device data" or "wearable device data" to keep the wording consistent.

Response:

We have revised the abstract for consistency, replacing "wearable" with "wearable device data."

Page 8: The section title "Initiatives for personalised population big biological data" sounds a

little awkward. Can the authors consider revising it such as "Population-scale initiatives for personalised big biological data"?

Response:

The section title has been revised to "Population-scale initiatives for personalised big biological data" as suggested.

Reviewer #2:

The paper provides a comprehensive overview of the integration of artificial intelligence (AI) and systems biology for interpreting multi-omics data in health and disease. It emphasizes the significance of longitudinal big biological data in advancing precision medicine, digital twins, and AI-driven healthcare. The authors discuss major technological advancements in multi-omics, computational modeling, and AI applications, highlighting their transformative potential in diagnostics, drug discovery, and clinical decision support.

It addresses the growing intersection of AI, systems biology, and precision medicine, aligning well with current research priorities and technological advancements. The discussion on digital twins is particularly valuable, as this concept is gaining traction in computational medicine. And it references a broad range of studies, providing substantial background for readers unfamiliar with the field.

Many of the AI and systems biology applications discussed have been covered extensively in previous literature. A more innovative perspective or unique analytical framework would enhance its impact.

Response:

We appreciate this suggestion. To address this, we have integrated a unifying analytical framework that connects the reviewed technologies to stages of the digital twin lifecycle—from data acquisition to simulation and intervention. This framework offers a forward-looking perspective on how multi-omics and AI tools converge in the context of personalized virtual models of health and disease.

“To move beyond summarising existing applications, we propose a forward-looking analytical framework emphasising the convergence of multi-scale modelling, longitudinal monitoring, and individualised prediction. This framework leverages digital twins, also defined as computational replicas of individuals that evolve with time, as dynamic vehicles to integrate systems biology models (e.g., GEMs and other biological networks), real-time data streams (e.g., wearable devices, EHRs), and AI-driven inference. This approach enables mechanistic insight, prospective prediction, and adaptive intervention by embedding causal modelling and continuous feedback loops. Furthermore, integrating explainable AI methods within this framework promotes interpretability and trust, which are critical for clinical adoption. This holistic, person-centric paradigm represents a shift from static snapshot analyses to dynamic, context-aware systems supporting proactive, precision healthcare and precision medicine.”

The paper provides an optimistic view of AI applications but does not sufficiently discuss the limitations of AI in big biological data analysis. Issues such as algorithmic bias, data heterogeneity, and the reproducibility of AI-generated predictions should be addressed.

Response:

We have added a new section entitled "Limitations and Risks of AI in Precision Medicine," and addressed the key challenges including algorithmic bias, lack of generalizability, data heterogeneity, and reproducibility issues. Strategies for mitigation, such as model validation and transparency, are also discussed

“While AI holds immense promise for advancing big biological data interpretation and precision medicine, it is essential to acknowledge its current limitations. Algorithmic bias remains a significant concern, especially when models are trained on datasets that underrepresent specific populations, leading to disparities in performance and potential inequities in clinical outcomes (Carini & Seyhan, 2024). Data heterogeneity, stemming from differences in experimental platforms, cohort characteristics, and data quality, can compromise model robustness and generalizability. Moreover, the reproducibility of AI-generated predictions is often hindered by opaque model architectures, a lack of standardised benchmarking, and limited access to code and training data. These challenges underscore the need for greater transparency, developing explainable AI methods, and establishing rigorous validation frameworks. By addressing these limitations, the field can ensure that AI applications in biology and medicine are robust, equitable, reliable, and clinically actionable (Kelly *et al*, 2019).

While AI offers powerful tools for data-driven healthcare, the risk of misinterpreting AI-generated outputs, especially in high-stakes clinical decision-making, must be carefully considered. Overreliance on AI predictions without appropriate clinical validation or understanding of model limitations can lead to diagnostic errors, inappropriate treatment choices, or missed adverse events. The complexity and opacity of many AI models, often called “black boxes,” exacerbate this risk by limiting interpretability and clinician trust. Therefore, integrating explainable AI techniques and fostering collaboration between AI developers and healthcare professionals is vital to ensure that AI tools complement rather than replace clinical judgment. Rigorous prospective validation, continuous monitoring, and clear communication of uncertainty are necessary to safeguard patient safety and optimise the clinical utility of AI-driven insights.”

The discussion on GEMs is well-developed, but the paper does not explore alternative computational models that might also contribute to multi-omics data integration. Including a comparative analysis of different computational approaches (e.g., deep learning models, graph-based AI models) would add value.

Response:

We have expanded the computational modeling section to include deep learning models (e.g., autoencoders, CNNs) and graph-based AI approaches (e.g., graph neural networks). We describe their use in omics integration and phenotype prediction, and we provide comparisons with GEMs where relevant.

“While biological networks offer a mechanistic and interpretable framework for simulating metabolic phenotypes, alternative computational models provide complementary strengths in multi-omics data integration. Deep learning models, such as variational autoencoders and multi-modal neural networks, excel at capturing complex, nonlinear patterns across diverse omics layers and have been used effectively for patient stratification, feature extraction, and disease prediction (Ballard *et al*, 2024; Ryu *et al*, 2018). Graph-based AI models, including graph neural networks (GNNs), are particularly suited for integrating biological networks (e.g., protein-protein interaction, gene regulatory, or signalling networks) with omics data,

enabling the modelling of topological dependencies and higher-order relationships (Valous *et al*, 2024). While often less interpretable than constraint-based models, these data-driven approaches offer scalability and flexibility, especially in high-dimensional, heterogeneous datasets. A comparative view that considers the trade-offs between mechanistic insight, predictive accuracy, and interpretability can guide the choice of computational tools for different research and clinical objectives in precision medicine.”

Given the paper's emphasis on AI-driven healthcare, a section on regulatory considerations (e.g., FDA approvals for AI tools, data privacy concerns, and ethical implications of digital twins) would be beneficial.

Response:

A new paragraph about the "Regulatory and Ethical Considerations in AI-Driven Healthcare" has been added in the conclusion. It covers topics including regulatory approval pathways (e.g., FDA clearance), patient privacy under frameworks like GDPR and HIPAA, and ethical concerns surrounding digital twin simulations and AI in clinical settings.

“As AI-driven tools become increasingly integrated into healthcare, addressing regulatory, privacy and ethical considerations is essential for responsible implementation. Regulatory frameworks, such as those from the U.S. Food and Drug Administration (FDA) and the European Medicines Agency (EMA), are beginning to adapt to the evaluation of AI-based clinical decision support systems, including requirements for transparency, robustness, and post-deployment monitoring (Palaniappan *et al*, 2024). Ensuring compliance with data privacy regulations like HIPAA and GDPR is especially critical when handling sensitive longitudinal big biological data. Using digital twins of individuals that continuously integrate multi-modal data raises novel ethical questions regarding informed consent, data ownership, and the potential for algorithmic determinism in health predictions (Xu *et al*, 2024). These concerns highlight the need for robust ethical frameworks, participatory design involving patients and clinicians, and governance structures prioritising fairness, explainability, and trustworthiness in AI systems. Addressing these dimensions may translate technological advances into equitable and clinically viable precision healthcare solutions.”

The potential risks of AI misinterpretation, particularly in high-stakes clinical decision-making, should be acknowledged.

Response:

We address this in the newly added limitations section, specifically noting the risks of AI-driven misinterpretation in clinical contexts. We emphasise the importance of human oversight, explainability, and rigorous validation in healthcare applications.

“While AI offers powerful tools for data-driven healthcare, the risk of misinterpreting AI-generated outputs, especially in high-stakes clinical decision-making, must be carefully considered. Overreliance on AI predictions without appropriate clinical validation or understanding of model limitations can lead to diagnostic errors, inappropriate treatment choices, or missed adverse events. The complexity and opacity of many AI models, often called “black boxes,” exacerbate this risk by limiting interpretability and clinician trust. Therefore, integrating explainable AI techniques and fostering collaboration between AI developers and healthcare professionals is vital to ensure that AI tools complement rather than replace clinical judgment. Rigorous prospective validation, continuous monitoring, and clear communication of uncertainty are necessary to safeguard patient safety and optimise the

clinical utility of AI-driven insights.”

This paper is highly relevant for researchers in computational biology, bioinformatics, and AI-driven healthcare. It serves as a useful review for those looking to understand the intersection of multi-omics data and AI applications.

We thank the reviewers once again for their detailed and thoughtful comments. These revisions have significantly strengthened the manuscript, both in terms of depth and clarity.

Sincerely,

Adil Mardinoglu
Professor of Systems Biology
adil.mardinoglu@kcl.ac.uk
adilm@scilifelab.se

Centre for Host-Microbiome Interactions
Faculty of Dentistry, Oral & Craniofacial Sciences
King's College London, UK

SciLifeLab, Science for Life Laboratory,
KTH-Royal Institute of Technology,
Stockholm, Sweden
<http://sysmedicine.com/>

30th Jun 2025

Manuscript Number: MSB-2025-12981R
Title: Longitudinal big biological data in the AI era
Author: Adil Mardinoglu
Hasan Turkez
Minho Shong
Vishnuvardhan Srinivasulu
Jens Nielsen
Bernhard Palsson
Leroy Hood
Mathias Uhlen

Dear Adil,

Thank you for submitting the revised version of your manuscript to Molecular Systems Biology. We have now received the enclosed reports from two reviewers who agreed to re-evaluate your work. As you will see below, both reviewers are overall satisfied with the revisions.

I am therefore pleased to inform you that we are able to accept your manuscript, pending the following minor amendments:

1. Please reduce the number of keywords to five.
2. Ensure consistency in the funding information between the manuscript file and the submission system. Specifically, the grant number 72254 for Knut and Alice Wallenberg Foundation is missing in the manuscript file, and the grant number National Research Foundation of Korea (NRF-2023-R1A2C3003438) is missing from the submission system.
3. Please remove the word count, number of figures, and number of references from the title page.
4. The section titled "Competing interest" should be renamed to "DISCLOSURE AND COMPETING INTERESTS STATEMENT".
5. While our designer can assist with redrawing the figures, it would be very helpful if you could provide specific instructions for each one-especially in light of the reviewers' comments regarding figure content. Please include a blurb outlining the specific changes you'd like to make to each figure, beyond general beautification or final touches. If possible, including a rough sketch would be even more helpful.

Click on the link below to submit your revised paper.

Kind regards,
Jingyi

Jingyi Hou, PhD
Senior Editor
Molecular Systems Biology

*** PLEASE NOTE *** As part of the EMBO Press transparent editorial process initiative (see our Editorial at <https://dx.doi.org/10.1038/msb.2010.72> , Molecular Systems Biology will publish online a Review Process File to accompany accepted manuscripts. When preparing your letter of response, please be aware that in the event of acceptance, your cover letter/point-by-point document will be included as part of this File, which will be available to the scientific community. More information about this initiative is available in our Instructions to Authors. If you have any questions about this initiative, please

contact the editorial office (msb@embo.org).

Reviewer #1:

Overall, the authors have addressed my comments well, except for the three related to the figures. I will leave these to the editor's discretion.

Reviewer #2:

This paper is appropriate for publication.

All editorial and formatting issues were resolved by the authors.

2nd Jul 2025

Dear Adil,

Thank you again for submitting your revised manuscript. I am pleased to inform you that your paper has been accepted for publication.

I have forwarded your figures to our designer, who will contact you directly regarding the redrawn images once they are ready.

After you approve the redrawn images, your manuscript will be exported to our production team to begin publication processing. It will undergo copy editing, and you will receive page proofs prior to publication.

Please note that you will be contacted by Springer Nature Author Services to complete licensing and payment information. When prompted by the Author Services system, please use the following token for payment: Token MTQ4NDAZMZYNA.

Should you be planning a Press Release on your article, please get in contact with embo_production@springernature.com after the manuscript is exported, in order to coordinate publication and release dates.

Thank you for your valuable contribution to Molecular Systems Biology!

Best wishes,
Jingyi

Jingyi Hou, PhD
Senior Editor
Molecular Systems Biology